# TRIM28-mediated nucleocapsid protein SUMOylation enhances SARS-CoV-2 virulence

Jiang Ren[1,11], Shuai Wang[2,11], Zhi Zong[3,11], Ting Pan[4,11], Sijia Liu[5], Wei Mao[6], Huizhe Huang[7], Xiaohua Yan[8], Bing Yang [3,9], Xin He [10] ✉, Fangfang Zhou [2] ✉ & Long Zhang [1,3] ✉

Viruses, as opportunistic intracellular parasites, hijack the cellular machinery of host cells to support their survival and propagation. Numerous viral proteins are subjected to host-mediated post-translational modifications. Here, we demonstrate that the SARS-CoV-2 nucleocapsid protein (SARS2-NP) is SUMOylated on the lysine 65 residue, which efficiently mediates SARS2-NP's ability in homo-oligomerization, RNA association, liquid-liquid phase separation (LLPS). Thereby the innate antiviral immune response is suppressed robustly. These roles can be achieved through intermolecular association between SUMO conjugation and a newly identified SUMO-interacting motif in SARS2-NP. Importantly, the widespread SARS2-NP R203K mutation gains a novel site of SUMOylation which further increases SARS2-NP's LLPS and immunosuppression. Notably, the SUMO E3 ligase TRIM28 is responsible for catalyzing SARS2-NP SUMOylation. An interfering peptide targeting the TRIM28 and SARS2-NP interaction was screened out to block SARS2-NP SUMOylation and LLPS, and consequently inhibit SARS-CoV-2 replication and rescue innate antiviral immunity. Collectively, these data support SARS2-NP SUMOylation is critical for SARS-CoV-2 virulence, and therefore provide a strategy to antagonize SARS-CoV-2.

The novel severe acute respiratory syndrome coronavirus 2 (SARS-CoV-2) contains a large single-stranded RNA genome with multiple open reading frames (ORFs) encoding for structural proteins: spike (S) glycoprotein, membrane (M) protein, envelope (E) protein,

nucleocapsid protein (NP; SARS2-NP), and 16 accessory proteins[1]. Similar to other coronaviruses, SARS2-NP electrostatically packages the positive strand viral genome RNA into a helical ribonucleoprotein (RNP) mainly due to an enrichment of highly positively charged amino

[1]The Eighth Affiliated Hospital, Sun Yat-sen University, Shenzhen 518033, China. [2]Institutes of Biology and Medical Sciences, Soochow University, Suzhou 215123, China. [3]MOE Key Laboratory of Biosystems Homeostasis & Protection and Innovation Center for Cell Signaling Network, Life Sciences Institute, Zhejiang University, Hangzhou 310058, China. [4]Shenzhen Key Laboratory of Systems Medicine for Inflammatory Diseases, School of Medicine, Shenzhen Campus of Sun Yat-sen University, Shenzhen 518107, China. [5]International Biomed-X Research Center, Second Affiliated Hospital of Zhejiang University, Zhejiang University School of Medicine, Hangzhou 310058, China. [6]Zhejiang Hospital, Zhejiang University School of Medicine, Hangzhou 310058, China. [7]Faculty of Basic Medical Sciences, Chongqing Medical University, Chongqing 400016, China. [8]Department of Biochemistry and Molecular Biology, School of Basic Medical Sciences, Nanchang University, Nanchang 330031, China. [9]Department of Pharmaceutical Chemistry and the Cardiovascular Research Institute, University of California, San Francisco, CA 94158, USA. [10]Institute of Human Virology, Zhongshan School of Medicine, Sun Yat-sen University, Guangzhou 510080, China. [11]These authors contributed equally: Jiang Ren, Shuai Wang, Zhi Zong, Ting Pan. ✉e-mail: hexin59@mail.sysu.edu.cn; zhoufangfang@suda.edu.cn; L_Zhang@zju.edu.cn

acid residues in the N-terminal domain and serine/arginine (SR)-rich linker region[2]. During the viral life cycle inside the host cells following SARS-CoV-2 infection, SARS2-NP interacts with the viral genome and M protein to facilitate the transcription efficiency of subgenomic viral RNA and promote assembly and release of mature virions[3]. In addition, SARS2-NP can quell the antiviral innate immune system through inhibiting retinoic acid-inducible gene I-like receptor pathway[4], phosphorylation and nuclear translocation of signal transducer and activator of transcription 1 and 2[5], and aggregation of mitochondrial antiviral signaling[6]. Moreover, SARS2-NP is extremely immunogenic, which enables it to elicit a protective immune response against SARS-CoV-2[7,8]. Therefore, diagnostic and therapeutic strategies have been developed to target SARS2-NP.

Intracellular liquid-liquid phase separation (LLPS) provides an efficient way to organize signaling molecules and compartmentalize bioreactions with high specificity[9,10]. In virus-infected cells, liquid droplets-like viral factories/inclusions or viroplasms formed by LLPS gather viral proteins, nucleic acids, and cellular factors to function as ideal hubs for viral replication and assembly. In addition, such viroplasms might also restrict the access of viral components to the cellular antiviral machineries and mitigate host innate immunity from activation[11–13]. Seminal studies conducted on single- or double-stranded RNA viruses have demonstrated that NP can drive the formation of phase-separated replication and transcription condensates in the host cytoplasm[14–17]. The structure of NP, which is well conserved among coronaviruses, shares several characteristics with proteins that undergo LLPS[6,18–20]. Thus, SARS2-NP itself forms biomolecular condensates, which was enhanced by viral genomic RNA[6,18–23]. Notably, the formed condensates can be interfered by small molecules, which serve as potential novel targets against the SARS-CoV-2 infection[6,18].

Increasing evidences indicate that coronavirus' proteins are subjected to different types of post-translational modifications (PTMs) mediated by the host enzymes, which has a remarkable effect on viral pathogenesis[24]. For example, palmitoylation and high glycosylation of SARS-CoV-2 S protein are crucial for viral infectivity[25,26]. Compared to unmodified SARS2-NP, protein-RNA interaction is reduced upon glycogen synthase kinase-3-mediated phosphorylation of the SR-rich linker, which leads to the formation of more dynamic liquid-like condensates for viral genome processing[3,23]. Similarly, we previously found the acetylation of SARS2-NP at lysine (K) 375 exhibits a weak ability of RNA binding and LLPS[6]. Here, we identify that SARS2-NP can be modified by SUMOylation. Previous study found that SARS1-NP undergoes SUMOylation, whereas the functional implications were not deeply revealed[27].

In this study, we demonstrate that SUMO conjugation, through associating with a SUMO-interacting motif (SIM) in SARS2-NP, is required for executing efficiently SARS2-NP's ability in homo-oligomerization, RNA association, LLPS, and innate antiviral immunosuppression. Moreover, an extra SUMOylation occurs at the site of the natural R203K mutant of SARS2-NP, which further enhances SARS2-NP function. We also identify that the SUMO E3 ligase tripartite motif containing 28 (TRIM28) conjugates SUMO to SARS2-NP. We further screened an interfering peptide that can disrupt TRIM28 and SARS2-NP interaction to rescue the innate antiviral response. Thus, our findings provide functional and mechanistic insights into the SARS2-NP SUMOylation. As a result, our study presents a novel intervention for SARS-CoV-2 infection.

## Results

### SARS2-NP is modified by poly-SUMO at K65

We first investigated the proteomic interactome of SARS2-NP in the SARS-CoV-2-infected A549 cells stably expressing human angiotensin converting enzyme 2 (hACE2) using tandem mass spectrometry (MS/MS). SARS2-NP-interacting proteins are listed in Supplementary Data 1.

Exactly as its crucial roles in the regulation of viral RNA activity, here, we also identified that SARS2-NP-interacting proteins are mainly involved in biological processes of RNA processing and gene expression using Gene Ontology (GO) enrichment analysis. A portion that associated with protein SUMOylation intrigued us, suggesting that SARS2-NP may undergo SUMOylation (Fig. 1a, Supplementary Fig. 1a). In mammals, the biochemical cascade of protein SUMOylation is sequentially catalyzed by the E1 activating enzyme, small ubiquitin-like modifier (SUMO)-activating enzyme subunit (SAE) 1/2 heterodimer, E2 conjugating enzyme, ubiquitin-conjugating enzyme 9 (Ubc9), and a diverse repertoire of E3 ligases which are usually required for attaching the C-terminal di-Glycine (G) motif of SUMO paralogue covalently to the ε-amino group of K residue within the substrate protein[28]. Ectopic expression of SARS2-NP and SUMO1/2/3 in HEK293T cells demonstrated the existence of SARS2-NP poly-SUMOylation, which was strongly enhanced by the SUMO-conjugating E2 enzyme, Ubc9 (Supplementary Fig. 1b). Furtherly, poly-SUMOylation of SARS2-NP was clearly observed in SARS-CoV-2-infected 16HBE, A549-hACE2 and CaCo-2 cells, as well as lungs of hACE2-transgenic mice (Fig. 1b), thereby confirming the existence of NP SUMOylation during SARS-CoV-2 infection.

To map the SUMOylation site, HEK293T cells were co-transfected with recombinant VSV-Flag-NP. SARS2-NP were then enriched by anti-Flag beads pull-down and subsequently digested by trypsin. Peptides containing GG-K were enriched by anti-K-ε-GG beads and analyzed by MS/MS (Supplementary Fig. 1c). Unlike ubiquitin, ISG15, NEDD8 which leaves a small GG remnant on the modified lysine residue after trypsin digestion, SUMO3 leaves a larger signature that severely hampers the identification of modified peptides. Whereas trypsin digestion of protein modified by SUMO3 T90R generates peptide with a GG remnant easily identifiable by classical MS/MS (Fig. 1c, left)[29–31]. In both SUMO3 WT-expressing cells and SUMO3 T90R-expressing cells, several GG-modified sites on SARS2-NP, i.e., K61, K266, K342, K347, K361, K375, K388, were identified by label-free quantification of the abundance of GG remnant. GG remnant on these sites are probably from endogenous ubiquitin, or ISG15, or NEDD8 modification. Only GG-modified K65 was identified specifically in SUMO3 T90R-expressing cells, indicating there is no endogenous GG remnant attached on K65, and K65 is the potential SUMOylation site of SARS2-NP (Fig. 1c, right, Supplementary Fig. 1d).

Substitution of the putative SUMOylation site K65 with an arginine (SARS2-NP K65R) completely blocked the SUMOylation of SARS2-NP in HEK293T, HeLa, Vero E6, RAW264.7 and MEF cells (Fig. 1d, Supplementary Fig. 1e). To visualize the intensity and subcellular location of SUMOylated SARS2-NP, we performed a proximity ligation assay (PLA), which allows for the detection of protein modification in situ with high specificity and sensitivity[32]. Using primary antibodies against Flag-SARS2-NP and HA-SUMO3, and secondary antibodies labeled with specific detection oligonucleotides, we observed that PLA signals that localized mainly in the cytoplasm. The SARS2-NP-SUMO3 PLA signal was undetectable in the SUMOylation-deficient mutants, K65R (Fig. 1e). In addition, the possibility that K65 ubiquitination was excluded by ubiquitination assay as necessary (Supplementary Fig. 1f). Thus, these observations confirmed K65 to be the SUMOylation site in SARS2-NP.

One of the most critical properties required by the NP for viral genome encapsidation is self-association. Thus, we investigated the effect of SUMOylation on SARS2-NP self-association using co-immunoprecipitation (co-IP) and found that SARS2-NP K65R led to less SARS2-NP self-interaction (Fig. 1f). Semi-denaturing detergent agarose gel electrophoresis (SDD-AGE) assay revealed more pronounced aggregation of SARS2-NP WT than that of SARS2-NP K65R (Fig. 1g). In addition, PLA showed that the SARS2-NP WT-SARS2-NP K65R together yielded much weaker signals in the cytoplasm compared to signals of SARS2-NP WT-SARS2-NP WT (Fig. 1h). Taken

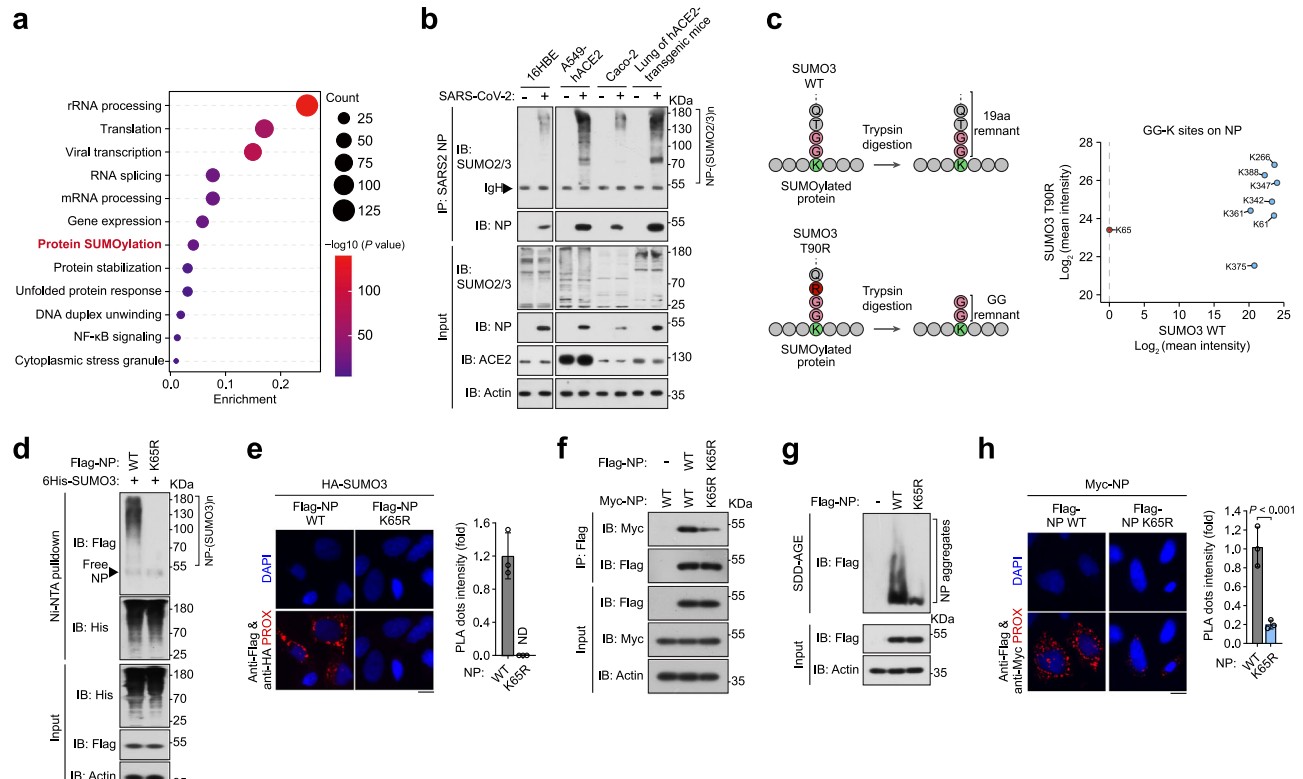

**Fig. 1 | SARS2-NP undergoes poly-SUMO modification. a** Scatter plot of enriched terms from gene ontology (GO) enrichment analysis of the SARS2-NP interactome in the SARS-CoV-2-infected A549 cells stably expressing hACE2 (A549-hACE2). **b** SUMOylation assay. Immunoblot (IB) of total lysates and anti-SARS2-NP immunoprecipitates (IP) of cell lysates from SARS-CoV-2-infected 16HBE, or A549-hACE2, or CaCo-2 cells, or lungs of hACE2-transgenic mice with SARS-CoV-2 infection. **c** Identification of NP SUMOylation site. Left, schematic representation of the signature tags left after trypsin digestion on peptides modified by SUMO3 WT/T90R. Right, the mean intensity (label-free quantification) of GG-K sites of SARS2-NP identified by MS/MS in SUMO3 WT/T90K-expressing HEK293T cells. **d** SUMOylation assay. IB of total lysates and Ni-NTA pulldown of cell lysates from HEK293T cells transfected with plasmids expressing 6His-SUMO3, plus Flag-SARS2-

NP WT/K65R plasmids as indicated. **e** In situ PLA for HA-SUMO3 and Flag-SARS2-NP WT/K65R in HeLa cells as indicated. Left, the PLA-detected proximity (PROX) complexes are represented by the red fluorescent dots. Right, changes in PROX dots intensity. NS, not significant; ND, not determined. **f** IB of total lysates and anti-Flag IP from HEK293T cells transfected with indicated expression plasmids. **g** Aggregation assay. SDD-AGE (top) and SDS-PAGE (bottom) of lysates of HEK293T cells transfected with indicated expression plasmids. **h** In situ PLA for Myc-SARS2-NP, plus Flag-SARS2-NP WT/K65R in HeLa cells as indicated. Right, changes in PROX dots intensity. Data are representative of at least three independent experiments with similar results (**a**–**h**). Data are presented as Mean ± SD; $n = 3$ independent samples; Statistical analyses were performed using a two-tailed Student's $t$-test (**h**). Scale bar, 10 μm (**e**, **h**).

---

together, these results show that SUMOylation ensures sufficient self-association of SARS2-NP.

## SUMOylation of SARS2-NP promotes its RNA association and LLPS

Given that SARS2-NP tends to undergo LLPS[6,18,19,23], we tested the effect of SARS2-NP SUMOylation on such a property. In this context, we generated a SUMOylation-mimicking fusion construct, called SUMO3-SARS2-NP, by putting SUMO3 before SARS2-NP. SARS2-NP and SUMO3-SARS2-NP were purified using a prokaryotic expression system (Supplementary Fig. 2a). We first did size exclusion chromatography to investigate purified proteins in solutions, As shown in Supplementary Fig. 2b, both SARS2-NP WT and SUMO3-SARS2-NP WT formed soluble and stable multimers. In comparison, SUMO3-SARS2-NP WT multimers were even larger in size as reflected in delayed eluting of major absorbance peaks and broader spectrum. Microscale thermophoresis (MST) assay, which was employed to quantify biophysical interactions between biomolecules in solution[33], showed that the binding capacity of SUMO3-SARS2-NP and SARS2-NP WT was higher than the self-interaction of SARS2-NP WT molecules (Fig. 2a). As SARS2-NP is essential for the assembly of RNP by packaging viral RNA, we evaluated whether the RNA-binding ability of SARS2-NP is also affected by SUMOylation. Result showed that the K65R substitution strongly inhibited the SARS2-NP RNA-binding activity (Supplementary

Fig. 2c). On the contrary, SUMO3-SARS2-NP had a higher RNA (labelled with Cyanine (Cy) 5) binding affinity than that of SARS2-NP WT as demonstrated by MST assay (Fig. 2c).

In droplets formation assay, pure EGFP- or AlexaFluor (AF) 488-labeled SARS2-NP demixed into spherical droplets in solutions with proper protein concentration which were not the artefacts of the EGFP tag (Supplementary Fig. 2d, Fig. 2d). Liquid droplets were obviously observed at weak acidic conditions (pH 4.5–7.5, Supplementary Fig. 2d) and low salt concentrations (50–150 mM, Supplementary Fig. 2e). The droplets formation was gradually reduced and finally diminished following pH or salt concentration increases (Supplementary Fig. 2d, e). Meanwhile, the LLPS ability of SARS2-NP was enhanced by RNA (Fig. 2d–f, i, Supplementary Fig. 2d, e). These observations are consistent with previous studies[18,21,23]. Interestingly, the EGFP-tagged SUMO3-SARS2-NP chimeric proteins formed larger droplets and were more abundant than the droplets formed by EGFP-SARS2-NP WT, without or with RNA (Supplementary Fig. 2d, e). To solidify this observation, we made a new version of SUMO3-SARS2-NP chimeric protein which could be dissociated effectively by TEV protease acting on the cut site in-between (Fig. 2c). As shown in Fig. 2d, dissociation of SUMO3-SARS2-NP resulted in a significant reduction of droplets formation to the level of SARS2-NP formed. Time-lapse microscopy was subsequently utilized to observe LLPS in real time, in which SARS2-NP WT was time-scale increased with the droplet size as expected (Fig. 2e).

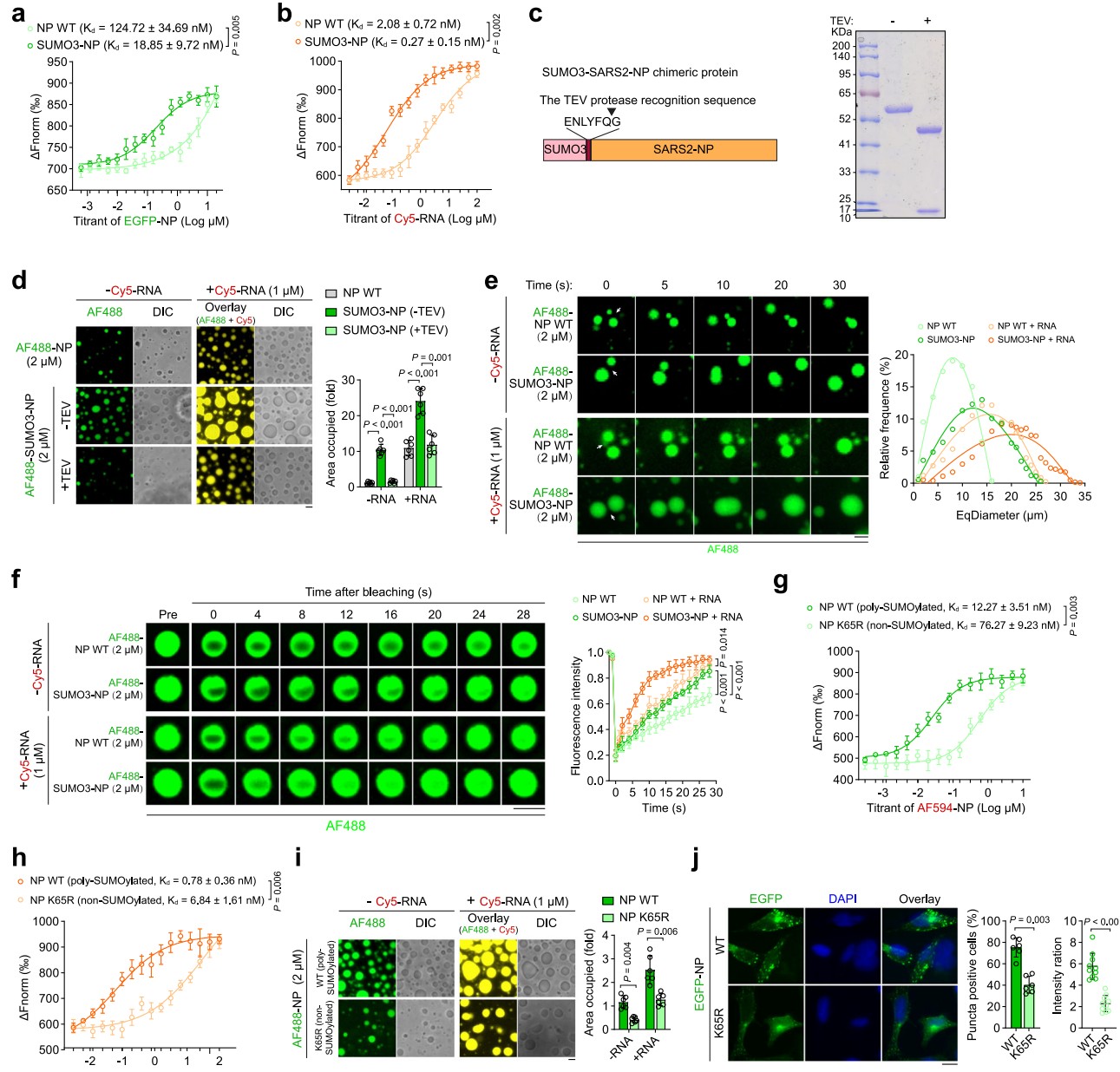

**Fig. 2 | LLPS ability of SARS2-NP can be enhanced by its poly-SUMOylation.** MST assay between ligand EGFP-SARS2-NP (**a**) or Cy5-RNA (**b**) and SARS2-NP or SUMO3-SARS2-NP. Data points indicate the difference in normalized fluorescence (Fnorm, ‰). **c** Schematic of SUMO3-SARS2-NP chimeric protein with a TEV protease cut site in-between (left). The cleavage result was analyzed by Coomassie-stained SDS-PAGE (right). **d** Droplet formation of AF488-labeled SARS2-NP WT, or SUMO3-SARS2-NP (without or with cleavage of TEV protease) without or with Cy5-RNA. Left, representative images. Right, fold change in droplet formation. **e** Fusion of AF488-labeled SARS2-NP WT, or SUMO3-SARS2-NP droplets without or with Cy5-RNA, at pH 5.5, 150 mM (left). And equivalent diameter (EqDiameter) frequency distribution of liquid droplets formed at 30 s (right). **f** FRAP assay of AF488-labeled SARS2-NP WT, or SUMO3-SARS2-NP droplets without or with Cy5-RNA. Left, representative images of before and after photobleaching. Right, quantification of FRAP over a 30 s time course. MST assay between ligand AF488-labeled SARS2-NP (**g**) or Cy5-RNA (**h**) and HEK293F cells-purified SARS2-NP WT/K65R. **i** Droplet formation of AF488-labeled SARS2-NP WT /K65R purified from HEK293F cells, without or with Cy5-RNA. Left, representative images. Right, fold change in droplet formation. **j** Puncta formation of EGFP-SARS2-NP WT/K65R in HeLa cells. Left, representative fluorescence images. Nuclei were visualized using DAPI staining. Middle, the percentages of cells harboring puncta in fluorescence positive cells. Right, the ratio of puncta-like fluorescence intensity. All data are representative of at least three independent experiments with similar results. Data are presented as Mean ± SD (**a**, **b**, **d**, **f**–**j**). $n = 3$ (**a**, **b**, **f**, **g**, **h**), or 6 (**d**, **i**, **j** middle), or 10 (**j** right) independent samples. Statistical analyses were performed using a One-way ANOVA (**d**, **i**, **j**), or Two-way ANOVA (**a**, **b**, **f**–**h**). Scale bar, 10 μm (**d**–**f**, **i**, **j**). DIC, differential interference contrast (**d**, **i**).

Strikingly, more rapid, abundant, and larger fusions were observed in SUMO3-SARS2-NP than those in SARS2-NP WT (Fig. 2e). Fluorescence recovery after photobleaching (FRAP) was used to assess the flow activity of SARS2-NP liquid droplets. After selectively bleaching the central part of the droplets, fluorescence recovery was observed within the SARS2-NP droplets, and more efficient diffusion was reflected within the SUMO3-SARS2-NP liquid droplets on a shorter time scale (Fig. 2f). Compared to SARS2-NP WT, SUMO3-SARS2-NP also exhibited a stronger LLPS ability with RNA (Supplementary Fig. 2d, e, Fig. 2d–f). As such, SUMO3 fusion and most likely SUMOylation promote SARS2-NP LLPS without or with RNA.

To purify poly-SUMOylated SARS2-NP from eukaryotic cells, we stably expressed Ubc9 and His-SUMO3 in HEK293F cells and transfected them with Flag-SARS2-NP WT/K0-R65K. Purification using

anti-Flag and Ni-NTA beads allowed us to produce a considerable amount of poly-SUMOylated SARS2-NP WT (Supplementary Fig. 2f). non-SUMOylated SARS2-NP K65R was also expressed in HEK293F cells and purified using anti-Flag beads. MST results indicated that poly-SUMOylated SARS2-NP WT had a stronger affinity for binding SARS2-NP (Fig. 2g) or RNA (Fig. 2h) than non-SUMOylated SARS2-NP K65R. We then compared the phase separation properties. Without or with RNA, poly-SUMOylated SARS2-NP WT rapidly formed micrometer-sized liquid droplets and quickly fused into larger ones, accompanied by increased fluorescence intensity; Whereas non-SUMOylated SARS2-NP K65R slowly formed much less and smaller droplets (Fig. 2i). *In cellulo*, HeLa cells ectopically expressing EGFP-SARS2-NP K65R displayed fewer and weaker cytoplasmic puncta (Fig. 2j). Putting together, these results demonstrated that the stronger SARS2-NP LLPS is bestowed by its SUMOylation.

## K65R mutation largely prevents SARS2-NP-caused innate immune evasion

To investigated the consequence of loss of SARS2-NP SUMOylation on the innate antiviral response, SARS2-NP WT/K65R expression plasmids along with interferon-β (IFN-β) or interferon-stimulated response element (ISRE) luciferase reporter were co-transfected in HEK293T cells, followed by Sendai virus (SeV) stimulation. Exogenous expression of SARS2-NP WT resulted in a significant downregulation of IFN-β and ISRE promoter activity, whereas SARS2-NP K65R largely abolished this inhibitory effect (Fig. 3a). The expression of *IFNB1* mRNA and downstream ISGs, *ISG56* and *CXCL10*, was barely inhibited by SARS2-NP K65R following SeV induction compared to that by SARS2-NP WT (Fig. 3b). Consistently, a slight inhibition of 5′-triphosphorylated RNA (5′-ppp RNA)- or poly(I:C)-induced *IFNB1* mRNA expression was also observed in SARS2-NP K65R (Fig. 3c). Enzyme-linked immunosorbent assay (ELISA) analysis indicated that the ectopic expression of SARS2-NP WT in RAW264.7 and A549 cells significantly repressed SeV-/VSV-induced IFN-β secretion, whereas SARS2-NP K65R expression had less effect (Fig. 3d). After challenging RAW264.7 cells with VSV, *Ifnb1* mRNA expression was inhibited, and the VSV titres (determined by plaque-forming; units, p.f.u.), VSV-specific mRNA (Fig. 3e), and glycoprotein (VSV-G; Fig. 3f) were significantly increased by SARS2-NP WT, but much less by SARS2-NP K65R. Accordingly, the K65R mutant strongly impaired SARS2-NP activity in elevating viral replication levels, as detected by VSV-GFP intensity in HEK293T cells (Fig. 3g), but not in type I IFN-deficient Vero E6 cells (Supplementary Fig. 3a). Furthermore, ectopic expression of SARS2-NP K65R mitigated the promotion of the SeV infection-induced cytopathic effect (CPE) compared to SARS2-NP WT (Fig. 3h). Therefore, K65R mutation, mostly due to loss of SARS2-NP SUMOylation, largely impairs SARS2-NP-mediated inhibition of innate immune response in vitro.

## SUMO conjugation and the SIM in SARS2-NP endow self-interaction

The results above raised the question of what is behind the poly-SUMO chain having such a higher capacity to elevate SARS2-NP LLPS. Substantial studies have announced that SUMO can selectively interact with SIM in a non-covalent manner[34–36]. In particular, the poly-SUMO:SIM pair could phase separate together[37]. We thus performed a sequence scan of SARS2-NP using the GPS-SUMO software and identified five potential SIMs that resemble the valine (V)/isoleucine (I)/leucine (L)-X-V/I/L-V/I/L motif (Fig. 4a). Then, three critical V/I/L residues were substituted with alanine (A), referred to as SARS2-NP SIM1/2/3/4/5 A, and we examined whether these SIMs were required for binding to SUMOylated SARS2-NP. Co-IP result showed that SARS2-NP SIM1A could not efficiently associate with NP WT (Supplementary Fig. 4a). Purified SARS2-NP WT directly associated with SUMO3-SARS2-NP chimeras. Whereas the SIM1A mutation reduced this affinity (Fig. 4b). MST assay also demonstrated that, compared to SARS2-NP

WT, SARS2-NP SIM1A mutant had a weaker affinity to bind poly-SUMOylated SARS2-NP (Fig. 4c) or SUMO3-SARS2-NP chimeras (Supplementary Fig. 4b). Moreover, the PLA signal was significantly weaker in SARS2-NP SIM1A and SARS2-NP WT co-transfected cells than in SARS2-NP WT and SARS2-NP WT co-transfected cells (Fig. 4d). In solutions, prokaryote-purified SUMO3-NP SIM1A formed multimers, but showed smaller size and narrower spectrum compared to SUMO3-SARS2-NP (Supplementary Fig. 2b). As well, poly-SUMOylated SARS2-NP SIM1A did not form aggregates as strongly as poly-SUMOylated SARS2-NP WT *in cellulo* (Supplementary Fig. 4c). Thus, SARS2-NP SIM1 interacts with the poly-SUMO chain of SUMOylated SARS2-NP.

To test whether the SIM1A mutation would alter the phase separation of SARS2-NP, we incubated AF594-labeled SUMO3-SARS2-NP chimeras with EGFP-tagged SARS2-NP WT/SIM1A. Compared to SARS2-NP WT, SARS2-NP SIM1A slowly formed a much smaller number of small liquid droplets with SUMO3-SARS2-NP (Fig. 4e). FRAP experiments showed that, after bleaching, SARS2-NP SIM1A with AF594-SUMO3-SARS2-NP was recovered less efficiently than SARS2-NP WT with AF594-SUMO3-SARS2-NP (Fig. 4f). Moreover, the SARS2-NP SIM1A mutation showed severely weakened ability of puncta formation *in cellulo* (Fig. 4g). To confirm these results, we compared the phase separation of AF488-labeled poly-SUMOylated SARS2-NP (eukaryote-purified) and SARS2-NP WT (prokaryote-purified) with that of poly-SUMOylated SARS2-NP and SARS2-NP SIM1A (prokaryote-purified). As shown in Fig. 4h, the combinations were made at various concentrations to generate concentration matrixes and corresponding phase diagrams. The SARS2-NP SIM1A mutant had compromised efficient for LLPS with poly-SUMOylated SARS2-NP. In another phase diagrams, lower competence to condense SARS2-NP WT (intact SIM1) as liquid-like droplets was seen apparently in eukaryote-purified SARS2-NP K65R (non-SUMOylated) compared with that in SARS2-NP WT (poly-SUMOylated) (Supplementary Fig. 4d). In the following FRAP experiments, interrupting poly-SUMO and SIM interaction-mediated SARS2-NP self-interaction, via loss of poly-SUMO modification or SIM site, resulted in lower phase reversibility and slower recovery rates of the formed droplets (Fig. 4i, Supplementary Fig. 4e). As predicted, the purified SARS2-NP SIM1A mutant exhibited a reduced binding affinity for viral RNA in oligo pull-down (Fig. 4j) and MST experiments in vitro (Fig. 4k). Correspondently, we observed that eukaryote-purified SARS2-NP SIM1A had compromised LLPS with viral RNA (Fig. 4l).

We thereafter examined the function of SARS2-NP SIM1A on the innate antiviral immunity *in cellulo*. In contrast to the SARS2-NP WT, SARS2-NP SIM1A exhibited a significantly reduced ability to suppress IFN-β promoter activity (Fig. 4m), *IFNB1* mRNA expression (Fig. 4n) and IFN-β secretion (Supplementary Fig. 4f), in addition to a reduced capacity for viral replication, as demonstrated by VSV-G expression (Supplementary Fig. 4g) and VSV-GFP intensity (Supplementary Fig. 4h). Furthermore, exogenous expression of SARS2-NP SIM1A resulted in decreased competence to elevate SeV-induced CPE (Supplementary Fig. 4i).

Our further experiments found that K65R-SIM1A double mutant (non-SUMOylated, and loss of SIM function) significantly reduced SARS2-NP's functions in self-association (Supplementary Fig. 4j), aggregation (Supplementary Fig. 4k), RNA binding (Supplementary Fig. 4m, n), droplets formation without or with RNA (Supplementary Fig. 4p), and suppression of innate antiviral signaling (Supplementary Fig. 4r), to a similar extent of SARS1-NP WT/SIM1A did. There were no additive or synergistic effects on reduction when comparing to what SARS2-NP SIM1A (poly-SUMOylated, but loss of SIM site), or SARS2-NP K65R (non-SUMOylated) did. Meanwhile, the similar consequences were shown in the corresponding double mutant mimicking protein NP SIM1A in solution purified from bacteria (Supplementary Fig. 2b, Supplementary Fig. 4l, o, q). Together, the intermolecular association between SUMO conjugation and SIM of SARS2-NP empowers

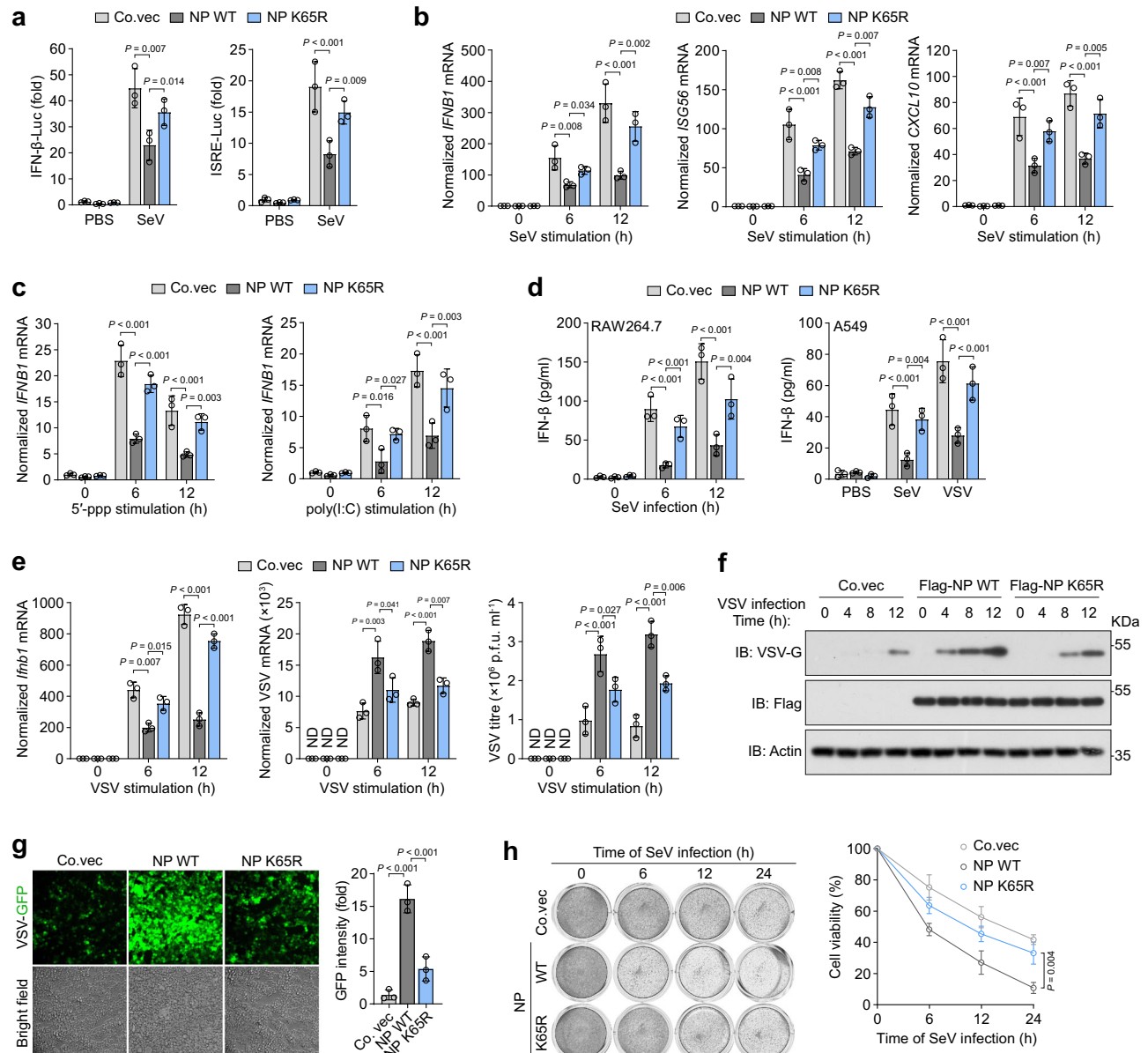

**Fig. 3 | K65R mutation largely recovers SARS2-NP-mitigated IFN-β signaling.**
**a** Fold change in IFN-β- and interferon-stimulated response element (ISRE)-luciferase (Luc) activity in HEK293T cells transfected with control vector (Co. vec), or indicated expression plasmids, followed by SeV infection for 12 h. **b** Normalized *IFNB1*, *CCL5* and *CXCL10* mRNA in HEK293T cells transfected with indicated expression plasmids, followed by SeV infection for the indicated times.
**c** Normalized *IFNB1* mRNA in HEK293T cells transfected with the indicated plasmids, followed by stimulation of 5′-ppp (left) or poly (I:C) (right) for 12 h.
**d** Determination of the IFN-β levels in RAW264.7 (left) and A549 (right) cells transfected with the indicated plasmids and infected with SeV for various time periods (left), and SeV/VSV for 12 h (right). **e** Fold change in *Ifnb1* mRNA (left) and VSV load (middle) as well as the VSV titres (right) in RAW264.7 cells transfected with indicated expression plasmids and infected with VSV (m.o.i. of 0.1) for indicated

times. p.f.u., plaque-forming units. ND, not determined. **f** Immunoblot (IB) of VSV glycoprotein (VSV-G) in MEF cells transfected with indicated expression plasmids and infected with VSV (m.o.i. of 0.1) for the indicated times. **g** Fluorescence microscopy and bright-field of VSV-GFP in HEK293T cells transfected with indicated expression plasmids, followed by infection with VSV-GFP (m.o.i. of 0.1) for 12 h (left). Scale bar, 100 μm. Right, fold change in VSV-GFP intensity. **h** SeV-induced cytopathic effect (CPE) in A549 cells after being transfected with indicated expression plasmids, followed by infection with SeV for the indicated time periods (left). Right, the rates of CPE. All data are representative of at least three independent experiments with similar results. Data are presented as Mean ± SD. *n* = 3 independent samples (**a**–**e**, **g**, **h**). Statistical analyses were performed using a One-way ANOVA (**a**–**e**, **g**) or Two-way ANOVA (**h**).

SARS2-NP itself LLPS or with RNA (Fig. 4o). Meanwhile, SARS2-NP SIM plays an important role in evading host innate antiviral immunity.

**SUMOylation and SIM of SARS2-NP are critical for viral infection**
To assess the role of SARS2-NP SUMOylation and SIM1 site in the host innate immune in vivo, as illustrated in Fig. 5a, we first generated recombinant VSVs (VSV, a prototype of RNA virus) by inserting the coding sequence of GFP, or SARS2-NP WT/K65R/SIM1A into the

virulence-attenuated VSV backbone between the glycoprotein (G protein) and polymerase protein (L protein)[38], which are denoted as VSV-GFP, VSV-NP WT/K65R/SIM1A respectively. Primary peritoneal macrophages were infected with recombinant VSVs at a multiplicity of infection (m.o.i.) of 0.1 for 6 or 12 h. VSV-NP WT dramatically promoted VSV replication, as reflected by elevated VSV titres and VSV-specific mRNA levels compared to control VSV-GFP. Whereas, both VSV-NP K65R and VSV-NP SIM1A showed attenuated abilities to promote VSV

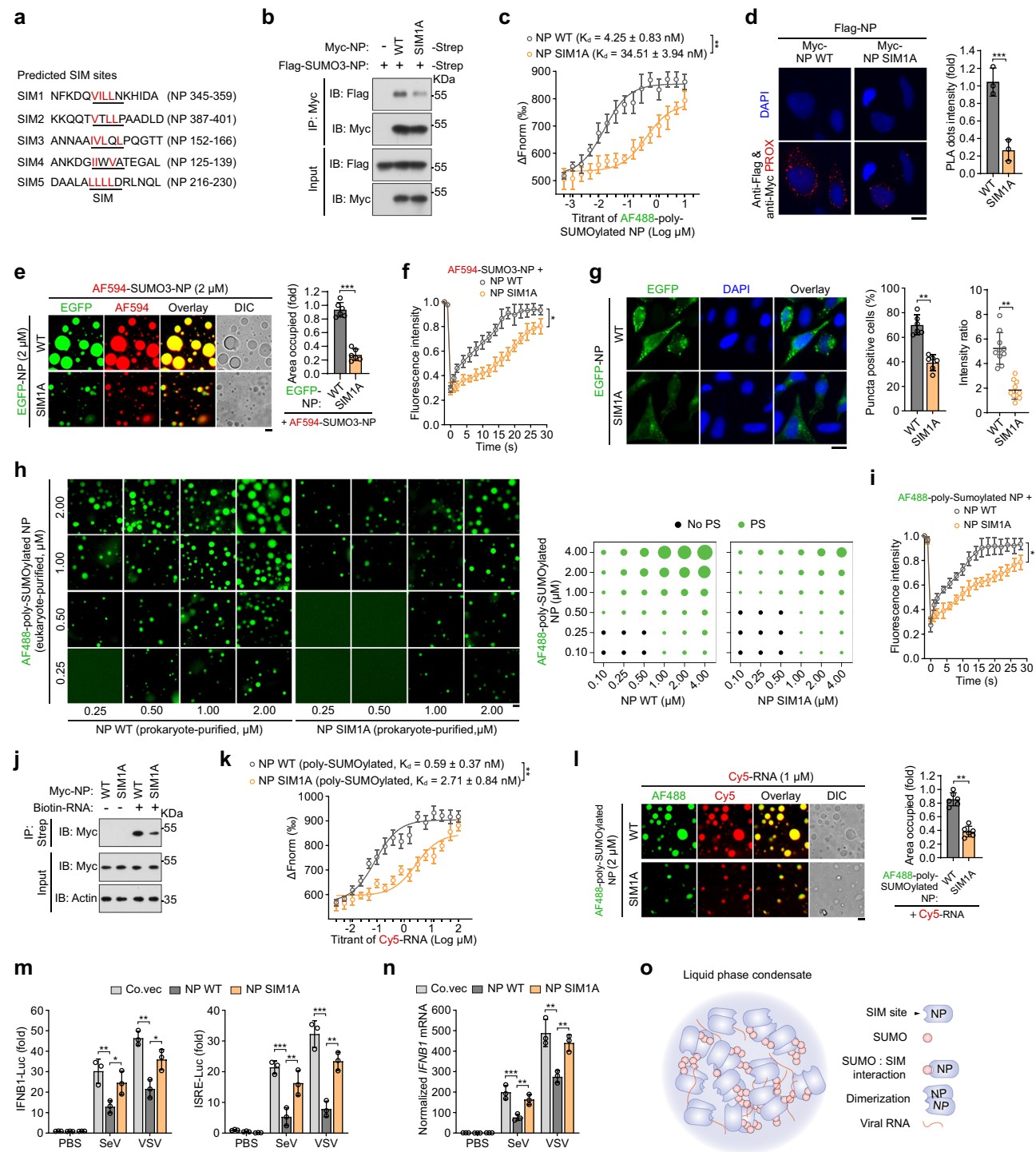

replication at similar levels (Fig. 5b). Furthermore, VSV-NP K65R and VSV-NP SIM1A did not inhibit *Ifnb1*, *Isg56*, and *Cxcl10* expression as efficiently as VSV-NP WT (Fig. 5c, Supplementary Fig. 5a). These results imply the importance of NP SUMOylation and SIM sites in NP-mediated viral propagation and innate immunosuppression.

Furthermore, we challenged mice with recombinant VSVs (Fig. 5d). The levels of VSV titres, VSV-specific mRNA (Fig. 5d), and VSV-G protein expression (Fig. 5e) were significantly higher in the spleen, liver, and lungs of VSV-NP-infected mice than in those of mice infected with control VSV-GFP. VSV replication in selected organs of VSV-NP K65R- and VSV-NP SIM1A-infected mice was also promoted, but to a much lesser extent. Correspondingly, repression of *Ifnb1*, *Isg56*, and *Cxcl10* mRNA in the spleen, liver, and lungs of VSV-NP K65R- and VSV-

NP SIM1A-infected mice was significantly weaker than that in VSV-NP-infected mice (Fig. 5f, Supplementary Fig. 5b). Besides, the IFN-β concentration in the sera was significantly higher in VSV-NP K65R- and VSV-NP SIM1A-infected mice than that in control VSV-NP-infected mice (Fig. 5g). We observed higher immune cells infiltration and severe tissue injury in the lungs of NP-VSV-infected mice than that in their control VSV-GFP-infected mice, as demonstrated by hematoxylin and eosin (H&E) staining. Whereas, mild manifestations were observed in the VSV-NP K65R- and VSV-NP SIM1A-infected counterparts (Fig. 5h). Consequently, the survival of mice infected with VSV-NP K65R/SIM1A was substantially improved (Fig. 5i).

To further determine the role of SARS2-NP SUMOylation and SIM in SARS-CoV-2 infection, Caco-2 cells were transfected to express

**Fig. 4 | SUMO-interacting motif (SIM) in SARS2-NP mediates multivalent self-interaction and thus enhances LLPS. a** Predicted SIM sites (underlined) using GPS-SUMO. SIM mutants by substitution of critical V, or L, or I residues with A (terms SIM1/2/3/4/5 A). **b** In vitro interaction of bacteria-purified Myc-SARS2-NP WT/SIM1A with Flag-SUMO3-SARS2-NP. **c** MST assay between ligand AF488-labelled SARS2-NP purified from HEK293F cells and bacteria-purified SARS2-NP WT/SIM1A. **d** In situ PLA for Flag-SARS2-NP and Myc-SARS2-NP WT/SIM1A in HeLa cells. **e** Droplet formation of EGFP-SARS2-NP WT/SIM1A with AF594-labeled SUMO3-SARS2-NP (left). All are bacteria-purified. Right, fold change in droplet formation. **f** FRAP assay of mixture of AF594-labeled SUMO3-SARS2-NP and SARS2-NP WT/SIM1A. All are bacteria-purified. **g** Puncta formation of EGFP-SARS2-NP WT/SIM1A in HeLa cells. **h** Droplet formation of AF488-labeled SARS2-NP purified from HEK293F cells, which mixed with bacteria-purified SARS2-NP/SARS2-NP SIM1A at indicated concentrations (left). Right, phase separation (PS) diagram. The green dots indicate PS, the black dots indicate no PS. **i** FRAP assay of mixture of AF488-labeled SARS2-NP purified from HEK293F cells and bacteria-purified SARS2-NP WT/SIM1A. **j** Immunoblot (IB) of total lysates and streptavidin RNA pull-down of lysates (IP) from HEK293T cells transfected with indicated plasmids. **k** MST assay between ligand Cy5-RNA and SARS2-NP WT/SIM1A purified from HEK293F cells. **l** Droplet formation of AF488-labeled SARS2-NP WT/SIM1A purified from HEK293F cells, which mixed with Cy5-RNA (left). Fold change in ISRE-Luc activity (**m**) and normalized *IFNB1* mRNA (**n**) in HEK293T cells transfected with indicated plasmids, followed by SeV/VSV infection for 12 h. **o** Schematic model of the SUMO-SIM multivalent interactions of SARS2-NP and their LLPS. Data are representative of at least three independent experiments with similar results (**b–n**). Data are presented as Mean ± SD (**c–g, i, k–n**). *n* = 3 (**c, d, f, i, k, m, n**), or 6 (**e, g** middle, **l**), or 10 (**g** right) independent samples. Statistical analyses were performed using a two-tailed Student's *t*-test (**d, e, g, i**), or One-way ANOVA (**m, n**), Two-way ANOVA (**c, f, k**). Scale bar, 10 µm (**d, e, g, h, i**). DIC, differential interference contrast microscopy (**e, i**).

SARS2-NP WT/K65R/SIM1A, and subsequently challenged with SARS-CoV-2. Compared to SARS2-NP WT, K65R or SIM1A mutant exhibited much less inhibition of *IFNB1* mRNA expression, resulting in a significant lower SARS-CoV-2 genomic RNA load (Supplementary Fig. 5c). Together, K65 SUMOylation and SIM1 of SARS2-NP could more efficiently suppress innate antiviral immunity.

## TRIM28 is identified as the E3 ligase for SARS2-NP SUMOylation

Next, we sought to chase down which SUMO E3 ligase mediates the SUMOylation of SARS2-NP. Analysis of the SARS2-NP interactome by MS revealed that TRIM28 is a unique SUMO E3 ligase that interacts with SARS2-NP (Fig. 6a, Supplementary Data 1). To confirm this possible interaction, we co-expressed SARS2-NP with several SUMO E3 ligases in HEK293T cells. Co-IP results showed that among the SUMO E3 ligases tested, only TRIM28 interacted with SARS2-NP (Supplementary Fig. 6a, b). Endogenous TRIM28 was also found to interact with SARS2-NP in HEK293T cells introduced with SARS2-NP (Supplementary Fig. 6c) or CaCo-2 cells following SARS-CoV-2 infection (Fig. 6b). PLA signals were detected by antibodies against TRIM28 and SARS2-NP, demonstrating their interaction inner the VSV-NP-infected HeLa cells (Fig. 6c). Subsequently, we studied the effect of TRIM28, through manipulating its expression, on SARS2-NP SUMOylation. Overexpression of TRIM28 promoted SARS2-NP poly-SUMOylation, whereas SARS2-NP poly-SUMOylation activity was reduced by the TRIM28 point mutant (C651A), with substitution of the cysteine residue (C) with A at position 651 which is required for its enzymatic activity (Fig. 6d). In HEK293T cells with *TRIM28* knockout (Supplementary Fig. 6d) or SUMOylation assay in vitro (Fig. 6e, Supplementary Fig. 6e), SARS2-NP SUMOylation was produced by introduction of TRIM28, rather than TRIM28 C651A. SUMOylation was still undetectable on SARS2-NP K65R when exogenous TRIM28 was introduced in HEK293T cells (Supplementary Fig. 6f) or in vitro (Fig. 6e), indicating that TRIM28 SUMOylates the K65 residue. Furthermore, significant attenuation of SARS2-NP poly-SUMOylation was observed following TRIM28 knockdown using shRNA #1/2 (Fig. 6f, Supplementary Fig. 6g). The PLA signals obtained from cells with TRIM28 knockdown verified significant lower levels of SARS2-NP SUMOylation than those from cells without TRIM28 knockdown (Supplementary Fig. 6h). So, these data support that TRIM28 is responsible for SARS2-NP SUMOylation.

We then studied the influence of TRIM28 on SARS2-NP LLPS. Both self-interaction (Supplementary Fig. 6i) and aggregation (Supplementary Fig. 6j) of SARS2-NP were increased upon ectopic expression of TRIM28, unlike inactive TRIM28 C651A. Whereas, TRIM28 knockdown decreased SARS2-NP aggregation (Fig. 6g). So, SARS2-NP from cells with TRIM28 knockdown could not form droplets with prokaryote-derived SARS2-NP as large and bright as those without TRIM28 knockdown (Fig. 6h). Simultaneously, FRAP assay also strongly suggested less efficient diffusion and lower reversibility within liquid droplets formed by SARS2-NP from TRIM28 knockdown cells with

prokaryote-derived SARS2-NP (Fig. 6i). Consistently, TRIM28 knockdown sharply hampered the formation of SARS2-NP-enriched puncta in living cells (Fig. 6j). In addition, oligo pull-down experiments demonstrated an obvious reduction in the binding affinity between SARS2-NP from TRIM28 knockdown cells and viral RNA (Supplementary Fig. 6k). Likewise, in comparison to SARS2-NP from intact cells, SARS2-NP from TRIM28 knockdown cells could not effectively condense into droplets with viral RNA (Fig. 6k). TRIM28 was not sequestered in the NP puncta or form puncta by itself upon co-expression of TRIM28 and NP in HeLa cells (Supplementary Fig. 6l). Therefore, combining with the evidences of TRIM28 WT/C651A overexpression (Supplementary Fig. 6m–q), we can conclude that TRIM28 promotes SARS2-NP LLPS.

## The widespread R203K mutation of SARS2-NP gains SUMOylation and increases LLPS and immunosuppression

The waves of the coronavirus disease 2019 (COVID-19) pandemic were propelled by the successive emergence of SARS-CoV-2 variants. Among them, the variants Alpha, Gamma, Delta, and Omicron were more prevalent and concerning, which can cause increased transmissibility, reduced effectiveness of vaccines or treatments, and more severe diseases. We noticed that these variants carry a R203K point mutation on SARS2-NP (https://cov-lineages.org/), which could potentially create a novel SUMOylation site predicted by SUMO-GPS. We then performed MS/MS to demonstrate the gain of K203 SUMOylation of SARS2-NP R203K mutant, as illustrated in Fig. 1c left and supplementary Fig. 1c. Result showed that GG remnant is attached on K203 of SARS2-NP R203K in SUMO3 T90R-expressing cells only (Supplementary Fig. 7a, b), in addition to K65. Additionally, SARS2-NP K65R-R203K mutant gained poly-SUMOylation based on K65R. The SARS2-NP R203K mutant, which contains two SUMOylation sites, showed higher poly-SUMOylation levels than SARS2-NP WT/K65R-R203K in an overexpression (Fig. 7a) or recombinant VSVs infection (Supplementary Fig. 7c) setting. The newly obtained poly-SUMOylation on SARS2-NP R203K could also be regulated by TRIM28 (Supplementary Fig. 7d, e). Compared to SARS2-NP WT/K65R, the ubiquitination level was not increased in NP R203K/K65R-R203K respectively, indicating there is lower level of or no ubiquitination happens on NP R203K (Supplementary Fig. 7f).

We then evaluated SARS2-NP LLPS upon gaining of an extra SUMOylation site. As demonstrated by the co-IP, SDD-AGE, and MST experiments respectively, SARS2-NP WT and SARS2-NP K65R-R203K had similar extend of self-association (Supplementary Fig. 7g), aggregation (Supplementary Fig. 7h), and binding affinity (Fig. 7b), all of which were presented much stronger by SARS2-NP R203K. With prokaryote-purified SARS2-NP WT, the droplets fused by eukaryote-purified SARS2-NP R203K had substantially larger size and stronger fluorescence intensity (Fig. 7c), higher phase reversibility, and faster recovery rates (Fig. 7d) than those by eukaryote-purified SARS2-NP

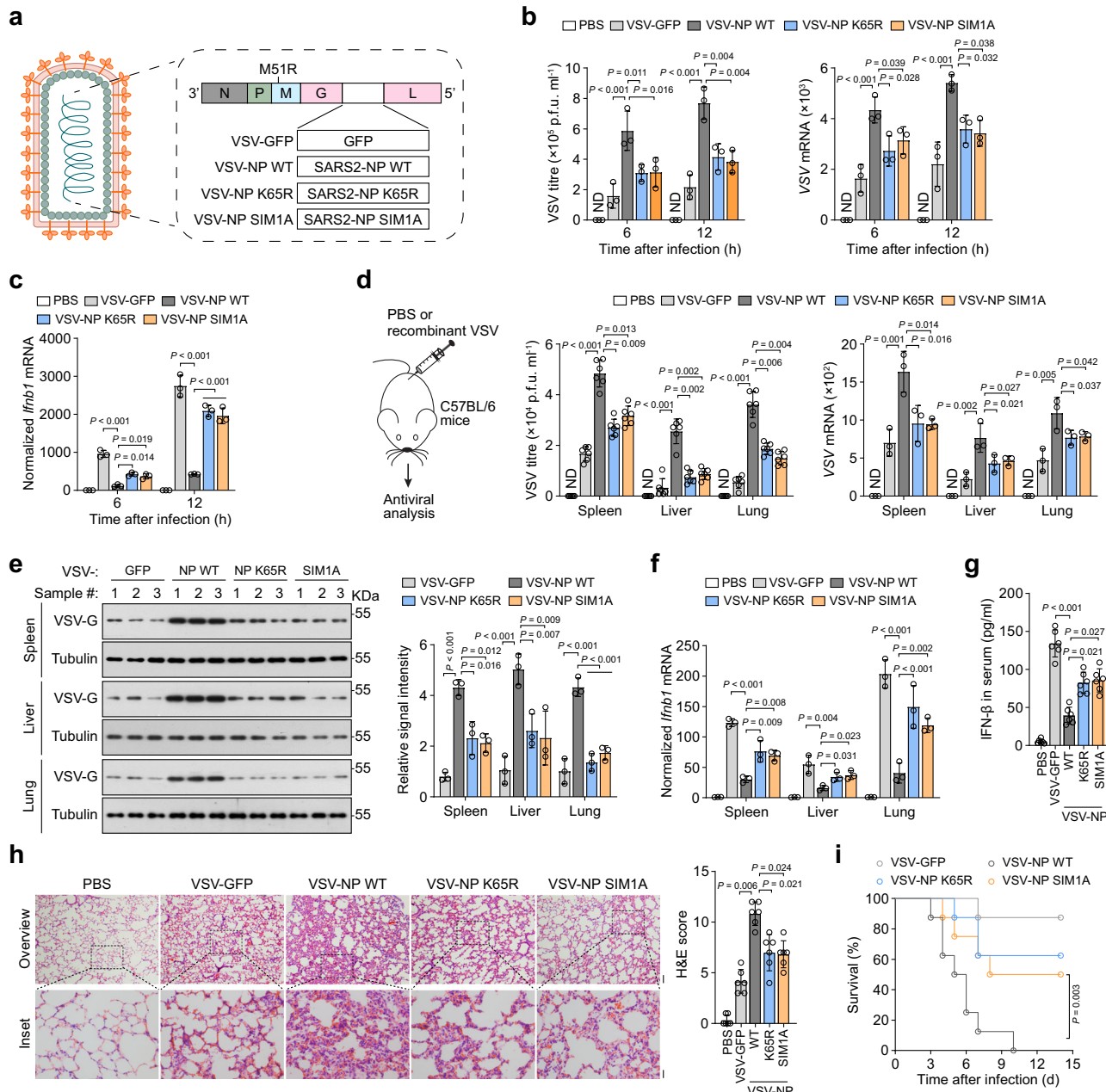

**Fig. 5 | Deprivation of SUMOylation or SIM largely rescues SARS2-NP-mediated innate immune suppression in vivo. a** Schematic of recombinant VSV harboring GFP, or SARS2-NP WT/K65R/SIM1A. **b** VSV titres (left, determined by plaque assay) and copy number (right) in mice peritoneal macrophages infected with indicated recombinant VSV at m.o.i. of 0.1, or not. **c** Normalized *Ifnb1* mRNA in mice peritoneal macrophages infected with indicated recombinant VSV at m.o.i. of 0.1, or not. **d** Recombinant VSV challenge strategy in C57BL/6 mice via intraperitoneal (i.p.) injection at $5 \times 10^8$ p.f.u. per mouse (6 mice each group). PBS was used as a negative control (left). VSV titres (middle) and copy number (right) in the spleen, liver, lungs of mice as indicated. **e** Immunoblot analysis of VSV-G in the spleen, liver, and lungs of the mice from **d** (left). Right, relative band intensity. **f** Normalized *Ifnb1*

mRNA in the spleen, liver, and lungs of the mice from **d**. **g** IFN-β concentration in the serum of the mice from **d**. **h** H&E staining of lung sections of mice from **d** (left). Right, the cumulative H&E score for quantification of lung lesions per mouse. Scale bars, 500 μm (left-top) and 100 μm (left-bottom). **i** Kaplan–Meier survival analysis of mice challenged with indicated recombinant VSV ($2 \times 10^9$ p.f.u. per mouse, i.p.). Data are representative of at least two (**d**–**i**) or three (**b**, **c**) independent experiments with similar results. Data are presented as the Mean ± SD (**b**–**h**). $n = 3$ independent samples (**b**, **c**) or mice (**d** right, **e**, **f**), or 6 independent mice (**d** middle, **g**–**i**). Statistical analyses were performed using a One-way ANOVA (**b**–**h**) or log-rank test (**i**). ND, not determined (**b**, **d**).

---

WT/K65R-R203K. Likewise, the condensation ability of SARS2-NP R203K was even higher in the cells (Fig. 7e). Furthermore, in contrast to SARS2-NP WT/K65R-R203K, SARS2-NP R203K showed an enhanced binding affinity (Supplementary Fig. 7i, j), and phase separation ability (Supplementary Fig. 7k) to/with viral RNA. In short, the SARS2-NP R203K mutation strengthens SARS2-NP LLPS.

As a result, the SARS2-NP R203K mutant was more efficient in suppressing SeV/VSV-induced IFN-β promoter activity (Supplementary

Fig. 7l), and *IFNB1* mRNA expression (Supplementary Fig. 7m), whereas these activities were repressed at lower levels by SARS2-NP WT/K65R-R203K. Then, we evaluated the functional consequences of the SARS2-NP R203K mutation. In that direction, recombinant VSVs carrying the SARS2-NP K65R-R203K, SARS2-NP R203K (hereafter, VSV-NP K65R-R203K and VSV-NP R203K, respectively) were engineered and applied to infect mice primary peritoneal macrophages or C57BL/6 mice. Similar to VSV-NP WT infection, VSV-NP K65R-R203K infection resulted

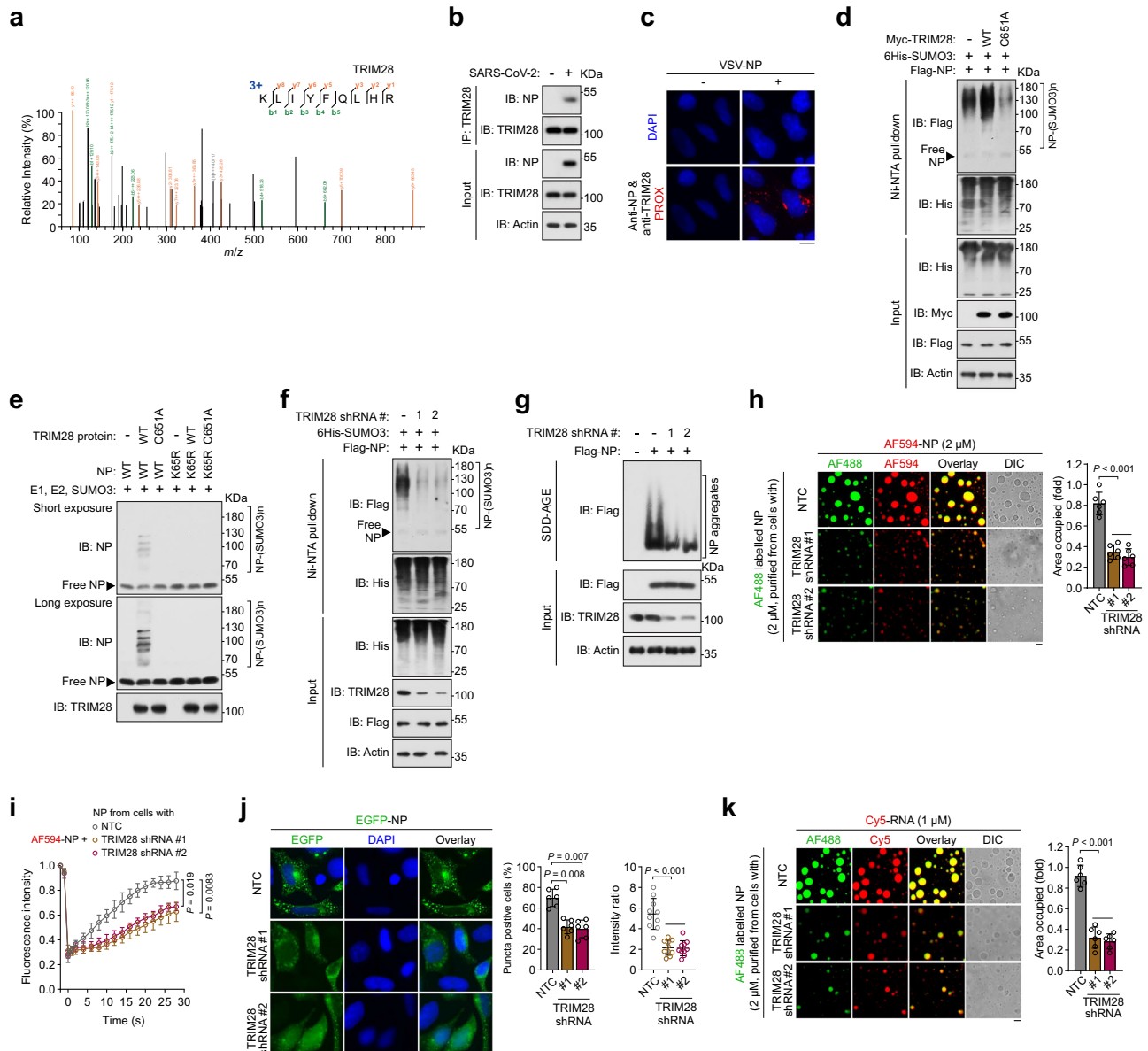

**Fig. 6 | The SUMO E3 ligase TRIM28 mediates SUMO conjugation to SARS2-NP.**
**a** Identification of TRIM28 as a potential SARS2-NP interactor using mass spectrum.
**b** Immunoblot (IB) of total lysates and anti-TRIM28 immunoprecipitates (IP) from CaCo-2 cells infected with SARS-CoV-2. **c** In situ PLA for TRIM28 and SARS2-NP in VSV-NP-infected HeLa cells. The PROX complexes are represented by the red fluorescent dots. **d**, **f** SUMOylation assay. IB of total lysates and Ni-NTA pulldown of cell lysates from HEK293T cells transfected with indicated expression plasmids. **e** SUMOylation assay in vitro. Prokaryote-purified proteins were incubated with SUMO E1, E2, SUMO3 as indicated. The results were analyzed by IB. **g** Aggregation assay. SDD-AGE (top) and SDS-PAGE (bottom) of lysates from HEK293T cells transfected with indicated expression plasmids. **h** Droplet formation of a mixture of AF594-labeled SARS-NP purified from bacteria and AF488-labeled SARS2-NP purified from HEK293F cells with NTC or TRIM28 shRNA. Right, fold change in droplet formation. **i** FRAP assay of a mixture of AF594-labeled SARS2-NP purified from bacteria and SARS2-NP purified from HEK293F cells with NTC or TRIM28 shRNA. **j** Puncta formation of EGFP-SARS2-NP in HeLa cells with NTC or TRIM28 shRNA. Left, representative fluorescence images. Middle, the percentages of cells harboring puncta. Right, the ratio of puncta-like fluorescence intensity to background. **k** Droplet formation of a mixture of Cy5-RNA and AF488-labeled SARS2-NP purified from HEK293F cells with NTC or TRIM28 shRNA. Right, fold change in droplet formation. All data are representative of at least three independent experiments with similar results (**b**–**k**). Data are presented as Mean ± SD (**h**–**k**). n = 3 (**i**), or 6 (**h**, **j** middle, **k**), or 10 (**j** right) independent samples. Statistical analyses were performed using a One-way ANOVA (**h**, **j**, **k**) or Two-way ANOVA (**i**). Scale bar, 10 µm (**c**, **h**, **j**, **k**). NTC, non targeting control (**h**–**k**). DIC, differential interference contrast microscopy (**h**, **k**).

in significant higher VSV titres, *VSV*-specific mRNA (Supplementary Fig. 7n, Fig. 7f), and VSV-G protein expression (Fig. 7g) in in vitro maintained macrophages or mice organs (spleen, liver, and lungs) than VSV-NP K65R infection. Not surprisingly, VSV replication was even more active in subjects infected with VSV-NP R203K (Supplementary Fig. 7n, Fig. 7f, g). Meanwhile, compared to VSV-NP K65R infection, the expression of *Ifnb1*, *Isg56*, and *Cxcl10* in macrophages maintained

in vitro (Supplementary Fig. 7o) or mice organs (spleen, liver, and lungs; Fig. 7h, Supplementary Fig. 7p), as well as the concentration of IFN-β in the mice sera (Fig. 7i), decreased accordingly to significant lower levels after VSV-NP K65R-R203K/R203K infection. In addition, VSV-NP R203K-infected mice had much stronger lung lesions (Fig. 7j), and a lower survival rate in a short time after infection (Fig. 7k) than the NP WT/K65R-R203K-infected mice. Thus, these functional studies

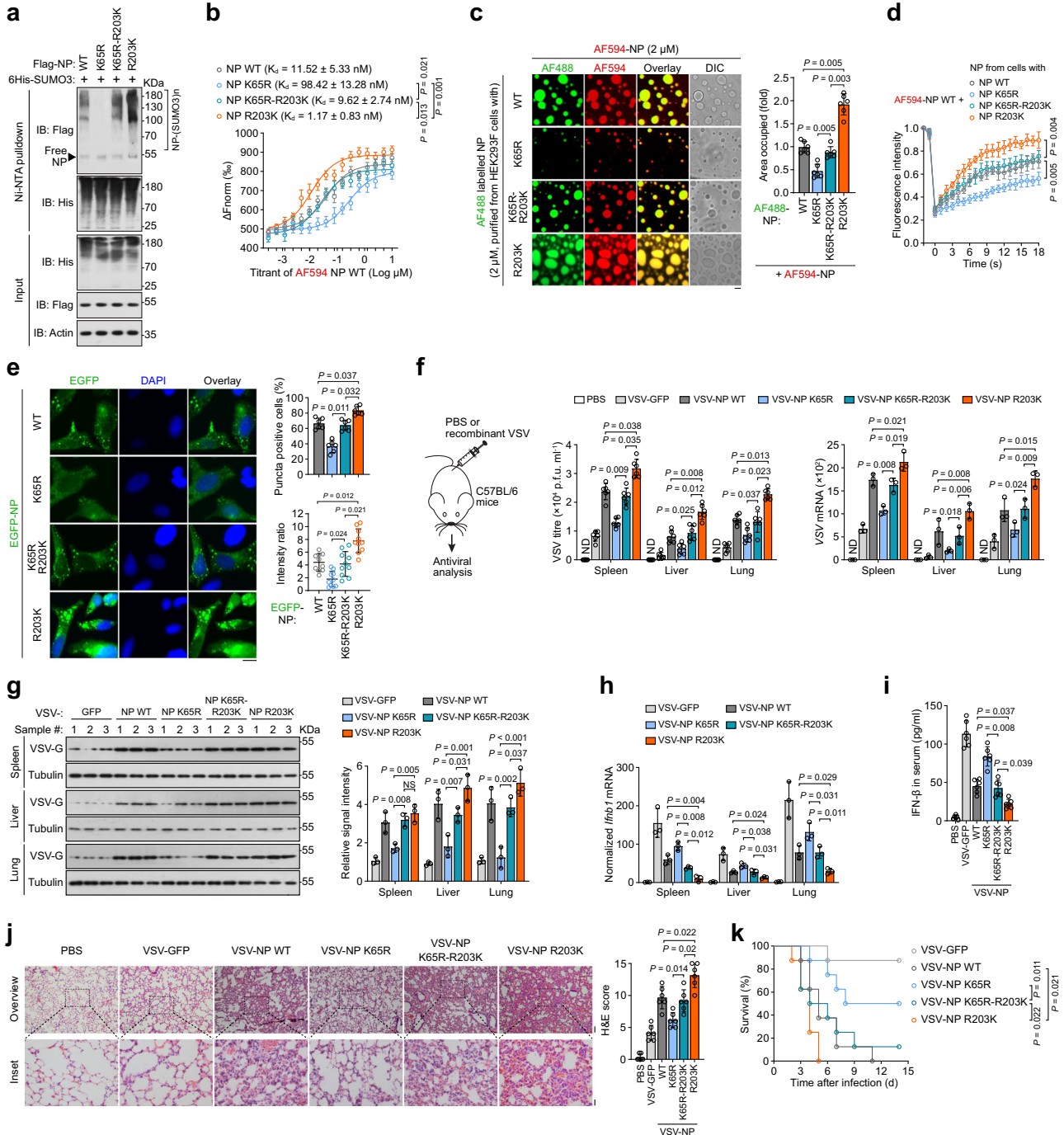

**Fig. 7 | Gain of SUMOylation at the site of natural R203K mutant of SARS2-NP further reduced the innate antiviral immunity. a** SUMOylation assay. Immunoblot (IB) of total lysates (input) and Ni-NTA pulldown of cell lysates from HEK293T cells transfected with indicated expression plasmids. **b** MST assay between ligand AF594-SARS2-NP (purified from bacteria) and SARS2-NP WT/K65R/K65R-R203K/R203K purified from HEK293F cells. **c** Droplet formation of a mixture of AF594-labeled SARS2-NP purified from bacteria and AF488-labeled SARS2-NP WT/K65R/K65R-R203K/R203K purified from HEK293F cells. Right, fold change in droplet formation. DIC, differential interference contrast microscopy. **d** FRAP assay of mixture of AF594-labeled SARS2-NP purified from bacteria and SARS2-NP WT/K65R/K65R-R203K/R203K purified from HEK293F cells. **e** Puncta formation of EGFP-SARS2-NP WT/K65R/K65R-R203K/R203K in HeLa cells. Left, representative fluorescence images. Right-top, the percentages of cells harboring puncta. Right-bottom, the ratio of puncta-like fluorescence intensity. **f** Recombinant VSV

challenge strategy in C57BL/6 mice (6 mice each group)via i.p. injection at $5 \times 10^8$ p.f.u. per mouse (left). Right, VSV titres and copy number in the spleen, liver, lungs. ND, not determined. **g** IB of VSV-G in the spleen, liver and lungs of the mice from **f**. Right, relative band intensity. **h** Normalized *Ifnb1* mRNA in the spleen, liver and lungs of the mice from **f**. **i** IFN-β concentration in the serum of the mice from **f**. **j** H&E staining of lung sections of mice from **f**. Right, the cumulative H&E score for quantification of lung lesions per mouse. **k** Kaplan–Meier survival analysis of mice challenged with indicated recombinant VSV ($2 \times 10^9$ p.f.u. per mouse, i.p.). Data are representative of at least two (**f**–**k**) or three (**a**–**e**) independent experiments with similar results. Data are presented as Mean ± SD (**b**–**j**). $n = 3$ (**b, d**) or 6 (**c, e** right-top) or 10 (**e** right-bottom) independent samples; 3 (**f** right, **g**) or 6 (**f** middle, **i, j, k**) mice. Statistical analyses were performed using a One-way ANOVA (**c, e, f**–**j**), or Two-way ANOVA (**b, d**), or log-rank test (**k**). Scale bar, 10 μm (**c, e**), 500 μm (**j** left-top), or 100 μm (**j** left-bottom).

imply that the SARS2-NP R203K mutant further inhibits innate antiviral immunity and heightens viral virulence owing to the gain of an extra SUMOylation site.

### Interfering peptide targeting TRIM28 and SARS2-NP interaction enhances the innate antiviral response by impairing SUMOylation and LLPS of SARS2-NP

Our experiments revealed a SARS2-NP-mediated decrease in the IFN-β promoter activity (Supplementary Fig. 8a), and *IFNB1* mRNA production (Fig. 8a) was significantly attenuated in the presence of TRIM28 shRNA after challenging with SeV/VSV. On the contrary, exogenous expression of TRIM28, rather than TRIM28 C651A, elevated SARS2-NP's inhibition potential (Supplementary Fig. 8b, c). In view of these results, we asked whether disrupting TRIM28 and SARS2-NP interaction could overcome SARS2-NP-mediated innate immune suppression. To do so, we first mapped the domains of TRIM28 and SARS2-NP, and generated domain deletions (Δ) to search the domains that determine their interaction (Supplementary Fig. 8d, e). Co-IP experiments showed that deletion of the coiled-coil (CC) domain of TRIM28 or the dimerization domain (DD) of SARS2-NP resulted in the disassociation of TRIM28 and SARS2-NP, indicating that these two domains are indispensable for TRIM28 and SARS2-NP interaction. By using a highly-integrated HDOCK platform, which can automatically incorporates the binding interface information from the PDB for protein–protein docking[39], a docking model between the TRIM28 CC domain (PDB: 6QU1) and SARS2-NP DD region (PDB: 6WJI) was characterized (Fig. 8b).

Next, we sought to interfere with TRIM28-SARS2-NP interaction and TRIM28-mediated SARS2-NP SUMOylation. According to the amino acid sequence of the TRIM28 CC domain, we designed and synthesized multiple interfering peptides, named NP SUMOylation interfering peptide (NSIP) I–V, with a D-retro-inverso conformation, where a reversed sequence of D-amino acids has almost the same structure, stability, and bioactivity as the parent L-peptides, but with more resistance to proteolytic degradation[40–42]. In parallel, a cell-penetrating peptide HIV-TAT was fused with NSIP for cellular delivery without energy consumption (Fig. 8c)[42,43]. NSIP-III was identified as a potent peptide that can interfere with the TRIM28-SARS2-NP interaction (Fig. 8d), SARS2-NP SUMOylation (Fig. 8e), as well as viral RNA binding ability of SARS2-NP (Supplementary Fig. 9a). Prokaryote-purified SARS2-NP protein was incubated with biotin-labelled NSIP-III. Streptavidin pull-down assay showed that NSIP-III can directly bind to SARS2-NP (Supplementary Fig. 9b). Thus, SARS2-NP in cells treated with NSIP-III showed remarkable less self-aggregation (Supplementary Fig. 9c) and puncta formation compared to the control (Fig. 8f).

We tested whether NSIP-III could counteract the inhibitory effect of SARS2-NP on innate immunity. C57BL/6 mice that had been pre-treated intraperitoneally (i.p.) with or without NSIP-III were infected with VSV-NP (Supplementary Fig. 9d). NSIP-III treatment significantly reduced the VSV titres, VSV-specific mRNA levels (Supplementary Fig. 9d), and VSV-G protein expression and in the spleen, liver, and lungs of the mice (Supplementary Fig. 9e). Compared to PBS-treated mice, the expression of *Ifnb1*, *Isg56*, and *Cxcl10* mRNA in the spleen, liver, and lungs (Supplementary Fig. 9f), and the concentration of IFN-β in the sera of the NSIP-III-treated mice was elevated considerably (Supplementary Fig. 9g). H&E staining confirmed that the lung damage caused by VSV-NP infection was alleviated by NSIP-III treatment (Supplementary Fig. 9h). Survival analysis demonstrated that NSIP-III treatment significantly increased the survival rate of the VSV-NP-infected mice (Supplementary Fig. 9i).

To evaluate the effects of NSIP-III treatment on the inhibition of SARS-CoV-2 replication, CaCo-2 cells were infected with SARS-CoV-2 and treated with NSIP-III. Following NSIP-III treatment, active viral replication was repressed gradually in a dose-dependent manner, as assessed by the expression of SARS-CoV-2 subgenomic RNA4-encoding *E* gene and viral titres. Correspondingly, *IFNB1* mRNA expression was elevated by NSIP-III treatment (Fig. 8g). While NSIP-III CaCo-2 cells were pretreated NSIP-I/II/IV/V is ineffective in antagonizing SARS-CoV-2 (Supplementary Fig. 9j). We further assessed NSIP-III treatment in the context of the SARS-CoV-2 infection in vivo. The hACE2 transgenic mice were pre-treated with PBS or NSIP-III i.p., and then intranasally inoculated with SARS-CoV-2 at a 50% tissue culture infectious dose (TCID$_{50}$) of $1 \times 10^5$. SARS-CoV-2 replication in the mice spleen, liver, and lungs was suppressed upon NSIP-III treatment (Fig. 8h). In addition, *Ifnb1*, *Isg56*, and *Cxcl10* mRNA levels in the spleen, liver, and lungs of NSIP-III-treated mice were significantly higher than those in PBS-treated mice (Supplementary Fig. 9k). Consistently, NSIP-III treatment promoted IFN-β secretion in the sera of SARS-CoV-2-infected mice (Fig. 8i). Immunohistochemical staining of spike proteins confirmed that SARS-CoV-2 replication was suppressed in the lungs of NSIP-III-treated mice (Fig. 8j). Furthermore, H&E staining indicated that SARS-CoV-2 infection led to severe lung damage in PBS-treated hACE2 transgenic mice, whereas NSIP-III treatment antagonized lung damage to a minimal level (Fig. 8k). Therefore, NSIP-III could counteract SARS2-NP LLPS and NP-induced suppression of innate antiviral immunity by interfering with TRIM28-mediated SARS2-NP SUMOylation (Fig. 8i).

## Discussion

As a living reservoir of viruses, cellular environment can be elaborately utilized by viruses to facilitate viral virulence and survival, including adopting a set of host cell machineries of trafficking, transcription, translation and PTMs, etc., and evading a variety of host immune response[44,45]. Uncovering the host-virus interactions is fundamental to address the potential scenarios of SARS-CoV-2 virulence and will offer critical insights into antiviral treatment[46]. PTMs have emerged as key molecular events behind host-virus interactions[24]. SUMOylation is an essential PTM process in all eukaryotes that controls diverse cellular events, such as signal transduction, localization, transcription, chromatin structure, and cell-cycle progression[47]. Following viral invasion into cells, the host SUMOylation machinery is sufficiently exploited by numerous viral proteins, including NS5 of ZIKA, IE1 of hCMV, E2 of HPV-16, and NS1 of influenza A, etc., to promote viral replication and assembly[48–51].

Here, we present evidence demonstrating that SARS2-NP is a bona fide SUMO substrate and occurs at the K65 residue, which is same as previous report[52]. A crucial step in the viral life cycle is packaging its genome into new virions. Self-association and homo-oligomerization are the main properties of the coronavirus SARS2-NP, which are essential for viral RNP formation and nucleocapsid assembly[3]. In this study, we found that SUMOylation is critical for SARS2-NP self-association and phase-separated condensates formation. Furthermore, SUMOylated SARS2-NP results to a stronger association and LLPS with viral RNA. Hence, SARS2-NP SUMOylation plays an important regulatory role in SARS-CoV-2 replication cycle and fitness.

Available evidences have supported that SARS2-NP LLPS is associated with its ability to inhibit IFN-β signaling for evading innate antiviral immunity[6,22]. In our findings, SARS2-NP SUMOylation is involved in this process. Depriving the SUMOylation of SARS2-NP results in increased IFN-β expression, lower viral propagation, and mice mortality, suggesting that SARS2-NP SUMOylation actively participates in maintaining viral virulence. Previous studies have indicated that the DD and IDR regions mediate SARS2-NP self-interaction, which is required for SARS2-NP LLPS[3,21]. Here, we identified a SIM in SARS2-NP and showed that SUMO conjugation can be recognized by this SIM. By utilizing intermolecular SUMO:SIM-established self-association, SARS2-NP is condensed into liquid-like droplets and high-molecule weight aggregates more efficiently. SUMOylation is a complicate system. There are myriads of proteins are subjected to SUMO modification *in cellulo*[28]. SUMOylated SARS2-NP could also interact with other

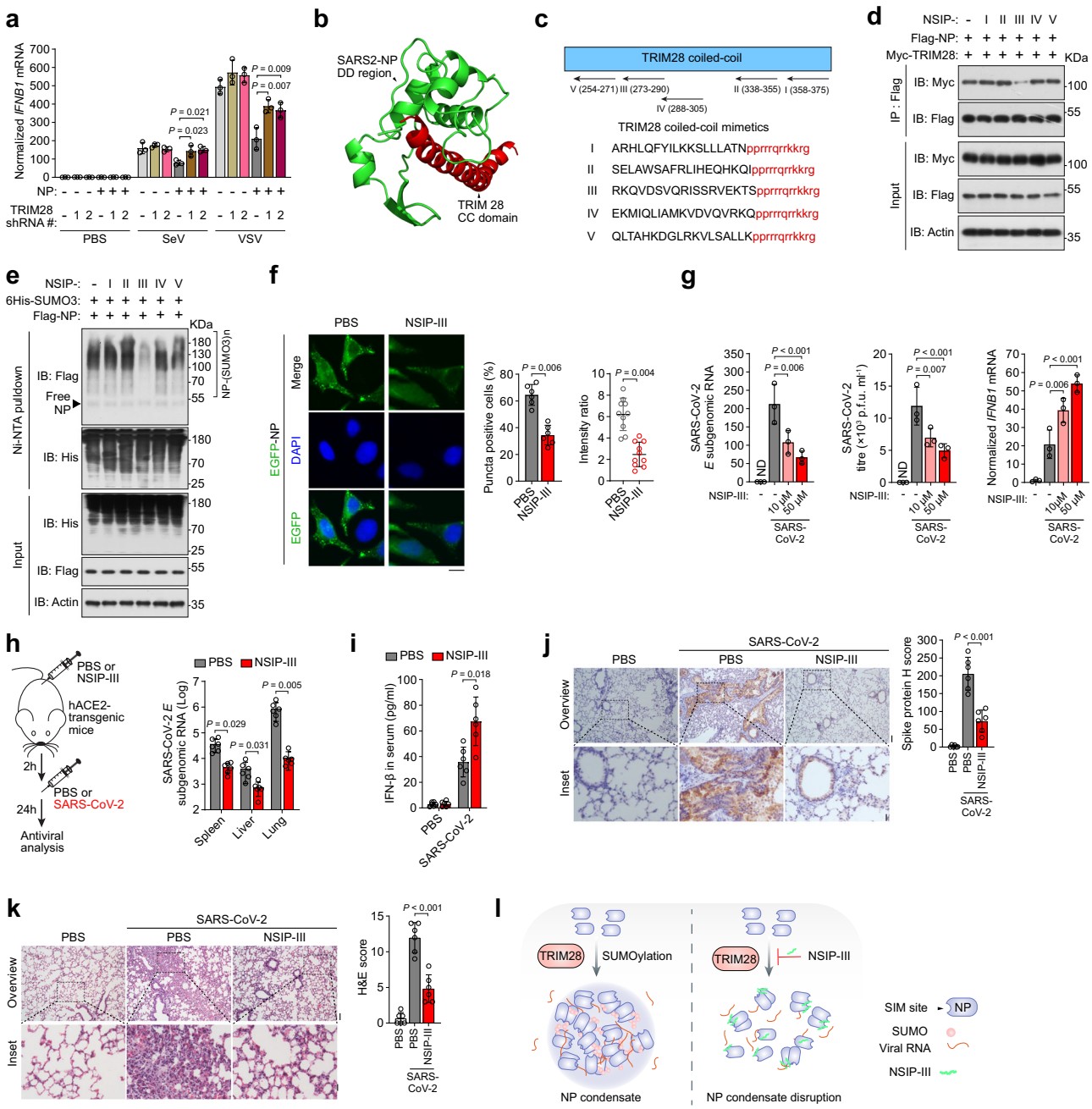

**Fig. 8 | The peptide NSIP-III disrupts SUMOylation and LLPS of SARS2-NP, thereby relieving immunosuppression. a** Normalized *IFNB1* mRNA in HEK293T cells transfected with indicated expression plasmids, followed by SeV/VSV infection for 12 h. **b** Predicted docking module of the TRIM28 CC domain (red) and SARS2-NP DD regions (green) interaction. **c** NSIP-I–V are designed to block SARS2-NP and TRIM28 interaction. Red, HIV-TAT cell-penetrating sequence. **d**, **e** Immunoblot (IB) of total lysates (input) and anti-Flag immunoprecipitates (IP, **d**) or Ni-NTA pulldown of cell lysates (**e**) from HEK293T cells transfected with indicated expression plasmids, followed by 10 μM of NSIP treatment for 12 h, or not. **f** Puncta formation of EGFP-SARS2-NP in HeLa cells treated with PBS or 10 μM of NSIP-III for 12 h. **g** CaCo-2 cells were pretreated PBS or NSIP-III for 2 h and infected with SARS-CoV-2 for 12 h. Left, the subgenomic RNA4-encoding *E* gene. Middle, the SARS-CoV-2 titres. Right, Normalized *IFNB1* mRNA. ND, not determined. **h** hACE2

transgenic mice were pretreated with PBS or NSIP-III (25 mg/kg) via i.p. injection for 2 h and inoculated with SARS-CoV-2 intranasally for another 24 h (left). Right, the subgenomic *E* gene in the spleen, liver and lungs. **i** IFN-β concentration in the serum of the mice from **h**. **j** Immunohistochemistry staining (left) and H score (right) of the SARS-CoV-2 S protein of lung sections of mice from **h**. **k** H&E staining of lung sections of mice from **h** (left). Right, the cumulative H&E score. **l** The peptide NSIP-III targeting TRIM28 and SARS2-NP interaction disrupts SUMOylation and LLPS of SARS2-NP, and relieves immunosuppression. Data are representative of at least two (**h**–**k**) or three (**a**, **d**–**g**) independent experiments with similar results. Data are presented as Mean ± SD (**a**, **f**–**k**). *n* = 3 (**a**, **g**) or 6 (**f**) or 10 (**f** right) independent samples, or 6 independent mice (**h**–**k**). Statistical analyses were performed using a One-way ANOVA (**a**, **g**, **j**, **k**) or two-tailed Student's *t*-test (**f**, **h**, **i**). Scale bar, 10 μm (**f**), 500 μm (**j** left-top, **k** left-top), or 100 μm (**j** left-bottom, **k** left-bottom).

SUMOylated or SIM-containing proteins. It is interesting to elucidate the possibility in the following study.

We noticed and verified that SARS1-NP can be SUMOylated on K62[27] (Supplementary Fig. 10a). The SIM site is conserve in SARS1 and 2-NP (Supplementary Fig. 10b). Meanwhile, we also observed the

association between SUMO conjugation and SIM of SARS1-NP promotes SARS1-NP itself LLPS or with RNA, as well as suppression of host innate antiviral signaling (Supplementary Fig. 10 c–i). Loss of (K62) SUMOylation or SIM site or both led to decreased SARS1-NP self-association (Supplementary Fig. 10c) and aggregation (Supplementary

Fig. 10d), weaker binding affinity to RNA (Supplementary Fig. 10e), compromised LLPS without or with RNA (Supplementary Fig. 10f–h), alleviated suppression of innate antiviral signaling (Supplementary Fig. 10i). Likewise, there was no additive or synergistic effect can be observed in loss of both (Supplementary Fig. 10). these results indicated that SUMO:SIM-mediated multivalent self-interaction of SARS-NP might be an extra and conserved and mechanism to enhance its functions.

To our surprise, an extra SUMOylation site was created on SARS2-NP R203K mutant, which exists in the Alpha, Gamma, Delta, and Omicron variants of SARS-CoV-2. The SARS2-NP R203K mutation has been demonstrated to promote viral replication efficiency and infectivity through an increase in phosphorylation of NP mediated by GSK-3β[53,54]. We also observed that SARS2-NP R203K gives rise to strengthened viral virulence due to the gain of an extra SUMOylation which intensifies the propensity of SARS2-NP in LLPS and IFN-β signaling inhibition. To address the possibility that increased SARS2-NP phosphorylation could affect the interpretation of our data regarding R203K mutation. We purified SARS2-NP R203K from eukaryotic cells treated with GSK-3β inhibitor Kenpaullone. R203K mutation can still increase LLPS of SARS2-NP significantly (Supplementary Fig. 7q). Furthermore, Kenpaullone only partially attenuated the inhibition of SARS2-NP R203K mutation on SeV/VSV-induced IFNB1 expression (Supplementary Fig. 7r). Therefore, SUMOylation occurs on the R203K is a novel mechanism underlying the enhanced viral virulence of SARS-CoV-2 R203K variant.

Our study uncovers an E3 ligase, TRIM28, that catalyzes SARS2-NP poly-SUMOylation. TRIM28, also known as KRAB-associated protein-1 (KAP1) or transcriptional intermediary factor 1β (TIF1β), mainly functions as a master corepressor of some transcription factors, especially zinc finger proteins (ZNFs). In addition, TRIM28 is well-characterized as a SUMO E3 ligase that can mediate the SUMOylation of PCNA, CDK9, and human adenovirus E1B-55K[55–57], etc. In our results, the corresponding changes in SARS2-NP properties in SUMOylation, self-association, LLPS, and innate immunity fluctuate with TRIM28 level or activity. Ubc9 can function as a SUMO-conjugating enzyme by directly recognizing and modifying the canonical ΨKxD/E motif of the target substrates. However, in some cases, the attachment of SUMO to the target protein requires a SUMO E3 ligase to confer specificity, especially to residues outside of the canonical motif[36,47]. The identification of TRIM28 as an E3 ligase for catalyzing SARS2-NP SUMOylation can explain why the SUMOylation sites of SARS2-NP, K65, and R203K, are not within the canonical motif.

Finally, we mapped the domains that mediated the interaction between SARS2-NP and TRIM28, i.e., the CC domain and DD region respectively. Based on this result, we designed interfering peptides to target the contact interface of SARS2-NP and TRIM28, and disrupt their interactions. A peptide named NSIP-III was determined to be a promising antagonist of SARS2-NP SUMOylation and LLPS for recovering the host antiviral immune response. NSIP-III was applied with relative high dose in this study. Thus, Loading NSIP-III with engineered ACE2-tropism nanoparticles would be more precisely and effectively to deliver NSIP-III in vivo.

In summary, our results offer novel insights and relevant consequences into the SUMOylation of SARS2-NP. A therapeutic strategy using interfering peptide is therefore provided to reduce SARS-CoV-2 virulence.

## Methods
### Cells
The HEK293T/F, HeLa, A549, Vero E6 and CaCo-2 cell lines were obtained from American Type Culture Collection. RAW264.7 cells were kindly provided by Stem Cell Bank, Chinese Academy of Sciences. Human bronchial epithelial cell line 16HBE was obtained from Sigma (SCC150). Mouse embryonic fibroblast (MEF) cells were isolated from embryo of C57BL/6 mice at E14.5 organogenesis. Peritoneal macrophages were harvested from the C57BL/6 mice 4 days after injection of thioglycolate (BD, 211716). Except 1640 medium for HEK293F cells culture, the other cells were cultured in DMEM medium supplemented with 10% fetal bovine serum (BSA, Gibco), 100 U/ml penicillin, and streptomycin in a humidified incubator at 37 °C with 5% $CO_2$. All the cells were checked for negative mycoplasma contamination at regular intervals.

### Plasmids and transfection
Eukaryotic expression plasmids for HA/6His-SUMO1/2/3, Myc/Flag/EGFP-SARS1/2-NP, and Myc-Ubc9, and reporter plasmids for IFNB1-Luc and ISRE-Luc have been described previously[6,58]. Sequences encoding SUMO E3 ligases, including CBX4, EGR2, KIAA1586, MUL1, NSMCE2, PIAS1, PIAS2α, PIAS2β, PIAS3, PIAS4, RNF212, TRIM1, TRIM22, TRIM27, TRIM28, TRIM32, TRIM38, TRIM39, ZBED1, and ZNF451 were amplified from cDNA library and inserted into pcDNA3.1-Myc/HA/mCherry vectors. Human ACE2 was inserted into pLV vectors. Prokaryotic expression proteins for (EGEP-) SARS1/2-NP, SUMO3-SARS1/2-NP and TRIM28 (C651A) were cloned into the pGEX vector with 2 × Strep at the C-terminus. Point mutations were generated via site-directed mutagenesis using KOD plus polymerase (Toyobo). shRNA oligos listed by Sigma (MISSION shRNA) for human TRIM28 knockdown were cloned into the PLKO.1 vector. Five shRNAs were tested, and the two most effective shRNAs, TRCN0000017998 (#1) and TRCN0000018001 (#2), were used for the experiments. CRISPR-mediated KO plasmids containing sgRNA (5′- CACCGATTGAGCTGGCAGTCTCGGC-3′) targeting human TRIM28 was generated in lentiCRISPR v2 (Addgene, 52961) according to the standard protocol. All plasmids were verified using Sanger sequencing.

Standard polyethylenimine (PEI 25000, Polysciences, 23966) was used for the transient transfection of plasmids into HEK293T cells. Plasmids were transfected into macrophages using the Geneporter 2 Transfection Reagent (Genlantis, T202007).

### Antibodies
The antibodies used in this study include: rabbit monoclonal anti-SUMO-2/3 (Cell Signaling, 4971; 1:1000 for IB), mouse monoclonal anti-SARS2-NP (ABclonal, A20142; 1:2000 for IB, 1:200 for PLA), rabbit monoclonal anti-SARS2-NP (Abcam, ab271180; 1:200 for IP), rabbit monoclonal anti-ACE2 (Abcam, ab108252; 1:1000 for IB), rabbit monoclonal anti-TRIM28 (ABclonal, A19568; 1:2000 for IB, 1:100 for IP/PLA), mouse monoclonal anti-Flag (M2) (Sigma, F3165; 1:2000 for IB, 1:200 for PLA), rabbit polyclonal anti-HA (Y11) (Santa Cruz, sc-805; 1:2000 for IB, 1:200 for PLA), rabbit monoclonal anti-HA (12CA5) (Santa Cruz, sc-57592; 1:2000 for IB), rabbit polyclonal anti-Myc (A-14) (Santa Cruz, sc-789; 1:2000 for IB, 1:200 for PLA), mouse monoclonal anti-Myc (9E10) (Santa Cruz, sc-40; 1:2000 for IB, 1:200 for IP), mouse monoclonal anti-His (H-3) (Santa Cruz, sc-8036; 1:2000 for IB), mouse monoclonal anti-β-actin (Sigma, A1978; 1:2000 for IB), rabbit monoclonal anti-β-tubulin (Cell Signaling, 2146; 1:2000 for IB), rabbit polyclonal anti-VSV-G (ABGENT, AP1016a; 1:1000 for IB), and HRP-conjugated secondary antibodies (Cell Signaling, 7076 (anti-mouse IgG) or 7074 (anti-rabbit IgG); 1:10,000 for IB).

### Viruses
To construct recombinant VSV carrying GFP, SARS2-NP WT, K65R, or SIM1A, the coding sequences were inserted into VSV backbone between the VSV glycoprotein (G protein) and polymerase protein (L protein) using NheI (Thermo Fisher, ER0972) and XhoI (Thermo Fisher, ER0691) restriction enzyme sites. The VSV backbone harbors an M51R substitution in the matrix (M) gene, attenuating the virulence of the original strain. The recombinant viruses (VSV-GFP, or VSV-NP WT/K65R/SIM1A/K65R-R203K/R203K) were recovered in Vero E6 cells.

Lentiviral particles were prepared by transfecting HEK293T cells with shRNA plasmids and the helper plasmids pCMV-VSVG, pMDLg-RRE (gag/pol), and pRSV-REV. The cell supernatants were harvested 48 h after transfection by filtering through 0.45 μm filters. Cells were infected with lentiviral supernatants at low confluence (20–40%) for 48 h. 2 μg/ml of Puromycin (Thermo Fisher, A1113803) was used to select and maintain stable cells.

A SARS-CoV-2 strain HB-01 used in this study was isolated from bronchoalveolar-lavage fluid samples of infected patient. The complete genome of this strain has been submitted to GISAID (accession ID:: BetaCoV/Wuhan/IVDC-HB-01/2020 | EPI_ISL_402119)[59]. The virus stocks were obtained from the supernatant of Vero E6 after inoculation for 48 h. The titers of viruses were determined by a standard $TCID_{50}$ assay on Vero E6 cell monolayers in 96-well plates with a 10-fold dilution series of the samples. All steps were performed within a biosafety level 3 (BSL3) facility.

## Mice
Six- to eight-week-old male C57BL/6 mice were purchased from Beijing Vital River Laboratory Animal Technology Co., Ltd, and maintained in a specific-pathogen-free facility at Soochow University. All animal experiments were approved and reviewed by the Institutional Committee for Animal Welfare of Soochow University (NO. SUDA20211211A01). The mice were infected with psuedovirus in the physical containment level 2 laboratory of Soochow University.

Six- to eight-week-old male hACE2 transgenic mice were purchased from Shanghai Model Organisms Center, Inc. All authentic SARS-CoV-2 challenge studies were approved by the Ethics Committee of ZSSOM of Sun Yat-sen University on Laboratory Animal Care (NO. SYSU-IACUC-2021-B0014) and conducted in a BSL3 facility and were performed in compliance with the guidelines and regulations of the Laboratory Monitoring Committee of Guangdong Province of China.

Mice were maintained under specific-pathogen-free conditions in the animal. The animal room has a controlled temperature (18–23 °C), humidity (40–60%), and a 12 light/12 dark cycle.

## CRISPR–Cas9-mediated gene knockout
HEK293T cells were infected with lentivirus for expressing Cas9 and *TRIM28* sgRNA. 48 h postinfection, 2 μg/ml of Puromycin was added to the medium as a primary screen of successfully transduced cells for another 48 h. Cells were then split and diluted to a concentration of 20 cells/10 ml, and seed into a 96-well plate for cloning. Single colonies were picked manually and verified for successful *TRIM28* knockout.

## Tandem mass spectrometry (MS/MS)
For identification of SARS2-NP proteomic interactome, total lysates were prepared from A549 cells stably expressing hACE2 infected with or without SARS-CoV-2. Non-infected cells were set as control. Immunoprecipitation (IP) with SARS2-NP antibody were then performed. Each condition was triplicated. Immunocomplexes were separated by SDS-PAGE and minimally stained with Coomassie brilliant blue. The entire lane was cut, digested with trypsin, eluted with 0.1% (v) TFA, and dried with SpeedVac. The peptides were analyzed on a Q Exactive HF-X Hybrid Quadrupole-Orbitrap Mass Spectrometer (Thermo Fisher) coupled with an Easy-nLC 1200 system. Spectral data were then searched against the reviewed UniProt *Homo sapiens* protein database including isoforms (released on April, 2021) by the MaxQuant software (v1.6.17.0). Mass tolerance values and the false discovery rate (FDR) were set at 20 ppm and 1% respectively. The maximum missed cleavage sites for trypsin was set to 2. Label-free protein quantification was switched on. Carbamidomethylation of cysteine residues was set as a fixed modification. While methionine oxidation, N-terminal acetylation were set as variable modifications. To avoid missing identification of the transient or weak interactions, the proteins presented by single peptide were retained. Statistical analysis was performed using Perseus (v1.6.1.3). To this aim, data (intensity) was $log_2$ transformed. Missing values were imputed with values representing a normal distribution with default settings in Perseus. To find statistically significant differences between the two conditions (infection *vs.* non-infection), a two-tailed Student's *t*-test with a permutation-based approach was applied with a FDR cut-off of 0.05.

For identification of SARS2-NP WT/R203K SUMOylation site, as illustrated in Fig. 1c left and Supplementary Fig. 1c, HEK293T cells were co-transfected with SUMO3 WT/T90K and Flag-NP WT/T90K for 36 h. SARS2-NP WT/T90K were then enriched by anti-Flag beads pull-down of lysates and subsequently digested by trypsin. Peptides containing GG-K were enriched by anti-K-ε-GG beads (Cell Signaling, 5562) and MS/MS were then performed. Spectral data were searched against the SARS2-NP WT sequence (UniProt: P0DTC9) or SARS2-NP R203K by MaxQuant. In addition to parameters set as above, GG (K) was added to variable modifications. GG (K) sites with a localization probability of >0.75 in at least 2 of the three biological replicates were retained.

## Gene Ontology (GO) enrichment analysis
GO enrichment analysis were conducted by the R package (cluster-Profiler, v3.17) to explore the biological process of SARS2-NP interactome with default parameters. Significant GO terms (false-discovery rate (FDR) < 0.05) were identified. GO enrichment results were further analyzed with DAVID for generating interactive graphs.

## Immunoblot (IB) and immunoprecipitation (IP)
To perform IB, the cells were lysed with RIPA buffer (20 mM Tris-Cl, pH 7.4, 150 mM NaCl, 1% Triton X-100, 0.5% sodium deoxycholate, 1% SDS, and protease inhibitor cocktail (Roche). Equal amounts of cells lysates were resolved in SDS-PAGE gel and transferred to polyvinylidene fluoride membrane (Millipore, IPVH00010). The membrane was incubated with appropriate primary and secondary antibodies (see below). The signals were visualized using chemiluminescence. The signal intensity of IB bands was quantified by Image J.

For IP, the cells were lysed with 1 ml IP lysis buffer (20 mM Tris-Cl, pH 7.4, 2 mM EDTA, 25 mM NaF, 1% Triton X-100, and protease inhibitor cocktail (Roche, 11697498001)) for 10 min at 4 °C. Equal amounts of lysate from each group were used. IP was performed with anti-Flag/Myc beads (Sigma, M8823/A7470) for 1 h or with various antibodies and protein G Sepharose (GE Healthcare, GE17061801) for 4 h at 4 °C. Thereafter, the beads were washed thrice with IP washing buffer (50 mM Tris-Cl, pH 8.0, 150 mM NaCl, 1% Triton X-100, and 0.5% sodium deoxycholate). The immunocomplexes were eluted using SDS sample buffer and boiled at 95 °C for 5 min. Standard IB was then performed.

## SUMOylation (or Ubiquitination) assay
Cells transfected with the indicated plasmids, or infected with SRAS-CoV-2 (m.o.i. of 0.1) were treated with 5 mM MG132 (SelleckChem, E2899) for 4 h prior to harvesting. After 36-48 h, the cells were washed with PBS containing 10 mM N-ethylmaleimide (NEM, Sigma, 128530) and lysed in two pellet volumes of RIPA buffer supplemented with protease inhibitor cocktail and 10 mM NEM. To detect SARS2-NP SUMOylation in the lungs of SRAS-CoV-2-infected hACE2-transgenic mice, 200 mg of lung was homogenized in 1000 μl of buffer 24 h after infection.

Under denaturing conditions, lysates were sonicated, boiled at 100 °C for 5 min, diluted with RIPA buffer containing 0.1% SDS, and centrifuged at 15,000 g for 15 min at 4 °C. Equal amounts of the lysate supernatants were incubated with anti-NP antibody (3 h) and protein G (1 h) orderly at 4 °C for detecting NP SUMOylation in SRAS-CoV-2-infected cells, or with anti-Flag beads for ubiquitination assay for 3 h at 4 °C. After extensive washing, the bound proteins were eluted with SDS sample buffer and subjected to IB analysis. For the Ni-NTA pulldown of 6His-tagged SUMOylated proteins, cells were resuspended in above

lysis buffer containing 8 M urea, and the 6His-tagged proteins recovered with Ni-NTA beads (Thermo Fisher, R90101) were eluted with a buffer containing 8 M urea and 20 mM imidazole.

### In vitro SUMOylation assay

The SUMOylation assay in vitro was carried out by using the SUMOylation assay kit following the manufacturer's instructions (Abcam). Prokaryote-purified recombinant proteins Flag-SARS-NP (K65R), without or with HA-TRIM28 (C651A), were incubated with SUMO E1 activating enzyme, and SUMO E2 conjugating enzyme Ubc9, SUMO3 at 37 °C for 1 h. The reaction was stopped by adding SDS-PAGE loading buffer, and the results were analyzed by IB.

### Semi-denaturing detergent agarose gel electrophoresis (SDD-AGE)

The cells were lysed in RIPA buffer (without SDS). After centrifugation at 15, 000 g for 30 min at 4 °C, the supernatants were resuspended in 1× sample buffer (0.5 × TBE, 10% glycerol, 2% SDS, and 0.0025% bromophenol blue) and loaded onto a vertical 1.5% agarose gel. After electrophoresis in the running buffer (1 × TBE with 0.1% SDS) at a constant voltage of 100 V at 4 °C, the proteins were transferred onto a PVDF membrane for IB.

### Streptavidin pull-down assay

To determine RNA binding to SARS2-NP in cell lysates, HEK293T cells were transfected with the indicated plasmids and incubated for 48 h. The cells were then washed twice with PBS, collected, and lysed in IP lysis buffer for 10 min at 4 °C. The cell lysates were centrifuged at 15,000 g for 15 min at 4 °C. 100 nM of biotin-labeled SARS-CoV-2 RNA was added to the pre-cleared lysates for 1 h incubation. then 15 μl of streptavidin Sepharose beads (Abcam, ab286845) were added to perform the standard pull-down assay.

To determine NSIP-III binding to SARS2-NP in solution, prokaryote-purified SARS2-NP protein was incubated with biotin-labeled NSIP-III (100 nM each) at 4 °C for 1 h. Streptavidin pull-down was then performed.

### Proximity ligation assay (PLA)

The cells were grown on collagen-coated Lab-Tek II chamber slides (Nunc, 154453), washed twice with PBS, and fixed in 4% formaldehyde in PBS for 15 min at room temperature. Subsequently, the slides were washed with TBS (25 mM Tris, pH 7.4, 100 mM NaCl), and permeabilized for 15 min in TBS containing 0.1% Triton X-100, and washed with TBST (0.05% Tween 20 in TBS). The slides were then blocked for 2 h with 0.5% milk powder in TBST and subsequently incubated with the appropriate combinations of antibodies overnight at 4 °C. After washing with TBST, proximity ligation was performed using Rabbit PLUS and Mouse MINUS Duolink in situ PLA kits (Sigma, DUO94102), according to the manufacturer's instructions. Subsequently, the slides were dehydrated, air-dried, and embedded in DAPI-containing anti-fadent mounting medium (VectorLabs, H-1200). PLA signals were imaged by a Zeiss LSM880 confocal microscope system. The PLA-detected proximity (PROX) complexes (red fluorescent dots) intensity per image (40 ×) was quantified by Image J with particle analysis tool.

### Microscale thermophoresis (MST) assay

Binding affinities between the target proteins and fluorescently labeled ligand proteins or ligand nucleic acids were measured in PBST binding buffer (0.05% Tween in PBS) using a MONOLITH NT.115 system (NanoTemper Technologies). Briefly, 10 of target protein was mixed with 10 μl of 2-fold serially diluted ligand at room temperature. The measurements were repeated on independent protein preparations to ensure reproducibility. The data were analyzed by plotting peptide concentrations against liquid-induced fluorescence changes (change in raw fluorescence on the Y-axis). Curve fitting was performed using

Prism 9 (GraphPad Software), and the $K_d$ values were calculated with a 95% confidence level.

### Protein purification

The corresponding prokaryotic proteins were generated in the *Escherichia coli* strain BL21. A single colony was picked and cultured overnight in LB medium supplemented with 50 μg/ml ampicillin. The overnight culture was diluted 1:100 with the same growth medium and was grown at 37 °C to an $OD_{600}$ of 0.6. The flask was left undisturbed at 16 °C for 30 min. Then, isopropyl-D-thiogalactopyranoside (IPTG, Thermo Fisher, 34060) was added to a final concentration of 0.5 mM and the culture was incubated overnight at 16 °C. The cells were harvested by centrifugation. The collected bacterial cell pellet was resuspended in lysis buffer (25 mM Tris-Cl, pH 7.4, 150 mM NaCl, 1 mM dithiothreitol (DTT), 1% Triton X-100, 100 μg/ml RNase, protease inhibitor cocktail, and 1 mM PMSF). RNase was added for preventing RNA prebound to NP, which was demonstrated by size exclusion chromatography as the value of A260 is less than half of A280 (Supplementary Fig. 2B). After sonication and freeze-thaw, the supernatant of the cell lysate was loaded onto a column containing Strep-Tactin Superflow beads (QIAGEN, 30004). The beads were then washed four times with lysis buffer. The purified proteins were eluted with lysis buffer.

For purification of eukaryote-expressed SARS2-NP, Flag-SARS2-NP WT/K65R/SIM1A plasmids were co-transfected into HEK293F cells with a stable expression of Ubc9 (neomycin selection) and His-SUMO3 (hygromycin B selection), with/without stable TRIM28 knockdown (puromycin selection). Following lysis in RIPA buffer (supplemented with protease inhibitor cocktail, 10 mM NEM, and 100 μg/ml RNase), sonication and freeze-thaw, the cell lysates were loaded onto a column containing anti-Flag beads. Bound proteins were eluted in lysis buffer containing 10 mM Flag peptide (Sigma, F3290). The elution from the column was pulled down with Ni-NTA beads and eluted using lysis buffer containing 250 mM imidazole. The purified proteins were dialyzed in buffer (25 mM Tris-Cl, pH 7.4, 150 mM NaCl, 1 mM DTT), and ensured via Coomassie-stained SDS-PAGE and stored at −80 °C.

### Size exclusion chromatography

500 μl of purified protein (3 mg/ml) was loaded onto a Superdex 200 increase 10/300 GL column (Cytiva). The column was pre-equilibrated with dialysis buffer (25 mM Tris, 150 mM NaCl pH 7.4) and calibrated using a set of molecular weight protein standards comprising bovine thyroglobulin (670 kDa), bovine gamma globulin (158 kDa), chicken ovalbumin (44 kDa) and horse myoglobulin (17 kDa). All size exclusion chromatography were carried out automatically using AKTA Prime Plus Liquid Chromatography System (GE Healthcare).

### Cleavage of SUMO3-SARS2-NP fusion protein with TEV protease

100 μM of SUMO3-SARS2-NP fusion protein with a TEV protease cut site in between SUMO3 and SARS2-NP was incubated with 0.2 IU/ml His-tagged TEV protease (GenScript) in buffer (25 mM Tris-Cl, pH 7.4, 150 mM NaCl, 1 mM DTT) at 4 °C overnight. Imidazole was then added into the protein solution at a final concentration of 10 mM. His-tagged TEV was removed using a column containing Ni-NTA. The cleavage result was analyzed by Coomassie-stained SDS-PAGE and stored at −80 °C.

### Protein labeling with fluorescent dye

All proteins were constructed with C-terminal cystine and conjugated with the maleimide derivative of Alexa fluorescent dye, according to the manufacturer's instructions. Briefly, 80 μM proteins were buffer displaced with labeling buffer (1× PBS, pH 7.4, 1 mM TCEP). maleimide Alexa Fluor (AF) 488/594 (Thermo Fisher, A10254/A10256) was added at a final concentration of 1 mM and incubated for 2 h at room

temperature. Excess dye was removed using a Zeba spin desalting column (Thermo Fisher, 89893) and stored at −80 °C until further use.

### In vitro phase separation assay

Purified proteins were diluted to indicated concentrations in buffer (25 mM Tris-HCl, pH 5.5, 150 mM NaCl, 1 mM DTT, unless specified otherwise) at room temperature. The protein solution (5 μl) was loaded onto a glass slide, covered with a coverslip, and imaged using a Zeiss LSM880 confocal microscope system.

Phase separation of recombinant SARS2-NP with Cy5-labeled SARS-CoV-2 RNA (5′-CACUCGCUAUGUCGAUAACAACUUCUGUGGCC CUGAUGGCUACCCUCUUGAGUGCAUUAAAGA-3′) was performed in the foregoing buffer. The mixture was transferred to a 96-well plate and imaged. The occupied area, or equivalent diameter (EqDiameter, the diameter of a circle with the same area as the measured object) of droplets per image (40 ×) were processed by Image J with particle analysis tool.

### Fluorescence recovery after photobleaching (FRAP) experiments

FRAP experiments were performed using a Zeiss LSM880 confocal microscope system. Spots of approximately 2 μm diameter in droplets of approximately 10 μm were photobleached with 20% laser power for 1 s using 488 nm lasers. Time-lapse images were acquired over 1 min after bleaching. Images were processed using the ImageJ, and the FRAP data were fitted to a single exponential model using the GraphPad Prism 9.

### Puncta observation *in cellulo*

The cells grown on collagen-coated Lab-Tek II chamber slides were transfected with plasmids expressing EGFP-tagged proteins for 24–36 h as indicated in figures, The cells were then washed with PBS, fixed with 4% paraformaldehyde in PBS for 15 min. The slides were dehydrated, air-dried, and embedded in DAPI-containing antifadent mounting medium. Images were collected using a Zeiss LSM880 confocal microscope system. The percentage of cells harboring puncta in fluorescence positive cells per image field (40 ×), and the ratio of puncta-like fluorescence intensity to background per cell were quantified by Image J.

### Enzyme-linked immunosorbent assay (ELISA)

IFN-β concentrations in the culture supernatants or serum were determined using ELISA kits (R&D Systems, QK410), according to the manufacturer's instructions.

### Luciferase reporter assay

HEK293T cells were seeded in 48-well plates and transfected with plasmids. Subsequently, cells were harvested for the measurement of luciferase activity after treatment, as indicated in the Figures. Co-transfection of Renilla was used to normalize the luciferase activity.

### Real-time quantitative PCR (qPCR)

Total RNA was isolated using RNAiso Plus (Takara, 9108Q). 1 μg of RNA was reverse-transcribed using the PrimeScript RT Reagent Kit (Takara, RR037B). qPCR was performed with TB Green Premix Ex Taq (Takara, RR82WR) using a StepOnePlus qPCR system (Applied Bioscience). The expression of all human/mouse target genes was normalized to that of the control gene *GAPDH/Gapdh*. The sequences of primers used are listed in Supplementary Table 1.

### Plaque assay

Mouse macrophages or other cell types ($2 \times 10^5$ cells) were plated 24 h before infection. The cells were infected with VSV at a m.o.i. of 0.1 or SeV (100 haemagglutination units (HAU)/ml) for varying times, as indicated in the figures. The VSV plaque assay and VSV replication were determined through a standard $TCID_{50}$ assay. After 1 h of infection, the plates were incubated for 48 h. The medium was then removed and the cells were fixed with 4% formaldehyde for 15 min and stained with 1% crystal violet for 30 min before plaque counting.

### In vivo viral infection

To perform in vivo viral infection studies, the C57BL/6 mice were infected with recombinant VSV ($5 \times 10^8$ p.f.u. per mouse) via intraperitoneal (i.p.) injection. The hACE2 transgenic mice were intranasally inoculated with SARS-CoV-2 HB-01 at a dosage of $1 \times 10^5$ $TCID_{50}$. Then, animals were euthanized and sampled at 24 h post inoculation. Blood was collected from the orbital sinus for ELISA. And the lungs, spleen, and liver from each mouse were collected for the analysis of RNA, protein, and viral titres.

To measure the VSV titres in the lungs, spleen, and liver, snap-frozen tissues were weighed and homogenized thrice for 5 s each in MEM. The suspensions were centrifuged at 1620 g for 30 min, and the supernatants were used for plaque assay on monolayers of Vero E6 cells. For survival experiments, the mice were monitored closely after VSV infection. The SARS-CoV-2 was quantified by qPCR analysis of SARS-CoV-2 *E* subgenomic RNA in the lungs, spleen, and liver via, and histological stain of spike protein in the lungs.

### Protein-protein interaction docking

TRIM28 CC domain (PDB: 6QU1) and SARS2-NP DD region (PDB: 6WJI) were selected as ligand and receptor respectively for protein-protein docking, using the HDOCK web service with default parameters (http://hdock.phys.hust.edu.cn/)[39]. Both ligand and receptor protein were prepared within the Protein Preparation Wizard in Schrodinger with default parameters. Exported PDB files were used for job submission.

### Interfering peptides

The NSIPs were as follows: NSIP-I, H-ARHLQFYILKKSLLLATNpprrr qrrkkrg-OH; NSIP-II, H-SELAWSAFRLIHEQHKQIpprrrqrrkkrg-OH; NSIP-III, H-RKQVDSVQRISSRVEKTSpprrrqrrkkrg-OH; NIP-IV, H-EKMIQ-LIAMKVDVQVRKQpprrrqrrkkrg-OH; and NIP-V, H-QLTAHKDGLRKV LSALLKpprrrqrrkkrg-OH. Lowercase font indicates the HIV-TAT sequence. The interfering peptides were synthesized by GL Biochem at >99% purity and stored at −20 °C in powder aliquots of 1 mg. The peptides were dissolved in PBS when using. For the in vivo experiments, the peptides were injected into mice via i.p.

### Histological staining

Mice lungs were dissected, fixed in 4% formalin, embedded in paraffin, sectioned into 5 μm thick, stained with hematoxylin and eosin solution (H&E), and the histological changes were imaged using light microscopy. The H&E score of lung lesions were assessed according to previous study[60]. The criterium is the extent of denatured and collapsed bronchiole epithelial cells, degeneration of alveoli pneumocytes, infiltration of inflammatory cells, edema, hemorrhage, exudation and expansion of parenchymal wall. Each item was scored for 0, normal; 1, mild; 2, moderate; 3, marked. The cumulative scores represent the total score per mice.

To visualize the spike protein, a primary antibody (ABclonal, A20022; 1:200) was used for immunohistochemical staining of the lung tissue sections from each group of mice. The quantification of staining was determined by an H score using the formula: 3 × percentage of strongly stained cells + 2 × percentage of moderately stained cells + percentage of weakly stained cells, yielding a range of 0–300.

### Statistics

Statistical analyses were performed using a two-tailed unpaired Student's *t*-test, or One/Two-way ANOVA with GraphPad Prism 9.0 as

indicated in the Fig. legends. All assays were performed at least three times. The exact value of n, which represents the number of mice used in the experiments, is indicated in the Fig. legends. For mice survival studies, Kaplan-Meier survival curves were generated and analyzed for statistical significance with GraphPad Prism 9.0. Pilot studies were used for estimation of the sample size to ensure adequate power. Statistical differences with a $P$ value of 0.05 or less were considered significant. No data points and mice were excluded from analysis. Details of statistical analyses and biological replicates are described in each figure legends.

## Reporting summary
Further information on research design is available in the Nature Portfolio Reporting Summary linked to this article.

## Data availability
The MS/MS raw data generated in the current study are available in the ProteomeXchange Consortium via the iProX partner repository with the dataset identifier PXD046366 (https://www.iprox.cn///page/project.html?id=IPX0007355000). All other data are provided in the article, Supplementary file. Source data are provided with this paper.

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

## Acknowledgements

The current work was supported by a special program from the Ministry of Science and Technology of China (2021YFA1101000, 2022YFA1105200), the Chinese National Natural Science Funds (U20A20393, 31925013, 32125016, 32070907, 32200575, 31771619, and 22074132), the Zhejiang Natural Science Fund (LD19C070001), Suzhou Medical College Basic Frontier Innovation Cross Project (YXY2303027), Suzhou Innovation and Entrepreneurship Leading Talent Program (ZXL2022505), Key Cross-research Projects of the School of Medicine, Soochow University (YXY2303027), Suzhou International Joint Laboratory for Diagnosis and Treatment of Brain Diseases, and Special Scientific and Technological Project for Emergency Treatment of Novel Coronavirus Infection with Traditional Chinese Medicine jointly funded by the National Administration of Traditional Chinese Medicine and Zhejiang Province (20230206). Additionally, we acknowledge the support from the platform of Sun Yat-sen University, Soochow University, Zhejiang University School of Medicine and the Life Sciences Institute.

## Author contributions

J.R. designed the experiments, analyzed the data and wrote the manuscript. J.R., S.W., Z.Z. S.L. and T.P. performed the experiments. X.Y., W.M., H.H., X.Y., B.Y., X.H. provided some experiments support. L.Z. and F.Z. provided valuable discussion.

## Competing interests

The authors declare no competing interests.
