## [Peer Review File · Nature Communications]

TRIM28-mediated nucleocapsid protein SUMOylation enhances SARS-CoV-2 virulenceEditorial Note: This manuscript has been previously reviewed at another journal that is not operating a transparent peer review scheme. This document only contains reviewer comments and rebuttal letters for versions considered at *Nature Communications*.

REVIEWER COMMENTS

Reviewer #3 (Remarks to the Author):

The authors have addressed my previous comments on their Nature Micro submission. I have no further comments.

Reviewer #4 (Remarks to the Author):

It appears that the authors have made a valiant effort in addressing the reviewer concerns by updating the manuscript and including additional control experiments. Nonetheless, I am concerned about major shortcomings that may have not been brought up or addressed in prior revisions.

Much of the premise of the paper relies on two critical mass spectrometry experiments: 1. The presumed discovery of a SUMO modification on a peptide containing NP-K65 (Supplementary data 2), and 2. A NP interactome dataset that identified the SUMO E3 protein TRIM28 (Supplementary Data 1). However, both these experiments produced puzzling results that are not exclusively consistent with a SUMO modification on K65:

1. For the identification of the SUMOylated NP peptide (line 114, Supplementary data 2), it is puzzling that the authors searched for di-G modified peptides because these indicate ubiquitination but not SUMOylation. The SUMOylation appendage for native SUMO would be much larger as there is no C-terminal Lys for trypsin cleavage. Hence, it is much more likely that K65 is significantly ubiquitinated (or ISGylated, which produces the same modification). Did the author check by Western blot for levels of ubiquitination or ISGylation? Both of those modification would leave the di-G appendage. It is important to clarify how Suppl. Data 2 was generated to identify the potential SUMOylation site? Was this NP overexpressed by itself, in which cell line? If in the presence of overexpression of a mutant SUMO form and/or E1/E2/E3 ligases, this needs to be described.

While the new mass spectrum in Supp. Fig. 1c showing the SUMOylated peptide is convincing, this was generated under overexpression conditions, and with mutant Q87R/Q88N SUMO, which adds a trypsin cleavage site to the C-terminus of SUMO. This spectrum still does not explain how the longer peptide (line 114) with di-G remnant could have been generated from a native SUMO modification.

2. The proteomics data in Supp. Data 1 needs further explanation. What is shown? Co-IP of NP? Where are non-transfected controls? How many replicates were analyzed and how were statistics calculated (none are presented). Several proteins, including TRIM28 are only identified by a single peptide. Standard in the field is to filter for at least 2 peptides per protein to ensure robust ID. The authors need to explain their rationale for less stringent IDs. Overall, it seems surprising that the authors focused on TRIM28 because only 1 peptide was IDed in the MS experiments. TRIM21, an E3 with ubiquitination activity, is identified much more confidently as an interactor (13 peptides/18 PSMs). Why did the authors not follow up on this lead and focused instead on TRIM28?

Even though subsequent experiments provide convincing evidence that K65 can be SUMOylated by TRIM28, all these experiments are carried out under conditions of overexpression of various components (E3, SUMO, etc.). Unfortunately, these results do not distinguish the possibility that another NP-K65 PTM may be the predominant proteoform in a native infection context. The only

experiment that tested for ubiquitination (Supplemental Figure 1e) used the NP-K0-R65K construct, which has all 29 other lysine residues mutated. As pointed out by reviewer 1, these mutations could influence NP stability and protein interactions, preventing the detection of native, more prominent PTMs on K65.

My subsequent concern is that the phenotypic experiments (interferon suppression, VSV infection) are all relying on the K65R NP as a non-SUMOylated form of NP. However, since the authors failed to convincingly show in the first sections of the paper that SUMOylation is the predominant PTM occurring on K65, I recommend a more cautious rewording of the data presented in Figures 3-5 & 7-8. Overall, the authors can conclude that the K65 (and R203K) NP modification sites (and SIM, etc.) are required for the observed phenotypes, but not necessarily SUMOylation.

Other comments:

Even with modifications and reorganization, the figures remain extremely dense and are at times challenging to follow.

When mapping the modifications on K203 in Supplementary Fig. 7a by mass spectrometry, the concern again is that this experiment relies on overexpression of the conjugating machinery and misses other potential di-G modifications that are also possible.

For new Supplementary Fig. 9b: Please clarify how this peptide binding assay was conducted. The IB shows the biotin signal at 55kDa, but the biotinylated peptide is likely much smaller. What is the signal here?

For the NSPI-III peptide experiments in mice and Caco2 cells, the more appropriate vehicle control would be the unmodified HIV-TAT sequence, or an inactive peptide. Did the authors test those for activity?

Experimental details are needed on the conditions and search settings for the mass spectrometry data analysis, in particular the search of modified peptides. Mass spectrometry raw data should be submitted to an online repository (e.g. PRIDE/ProteomeXchange).

How do interactors compare to other NP interaction datasets, e.g. Gordon, D.E. et al. Nature 583, 459–468 (2020)?

Explain how GO analysis in Fig. 1 and Supp. Fig. 1a differ. What do the arrows represent?

Line 201: HeLa data is shown in Fig. 2k - not 2e. Why the switch to HeLa cells here? Clarify whether these cells also overexpressed Ubc9 and SUMO3.

Line 240: Please explain the MST assays.

In Supp. Fig. 6a in the input, NP is below the 35kDa MW marker, while in the IPs it appears slightly below the 55kDa marker. Is this an error?

Supplemental Fig. 8e: The MW marker shows that myc-TRIM28 is <55kDa, but the full-length protein is ~100kDa (see Fig. 8d). Is the MW marker incorrect, or was a TRIM28 truncation used for these experiments?

Line 441: Explain the HDock platform.

Line 451: incomplete sentence

Reviewer #3 (Remarks to the Author):

The authors have addressed my previous comments on their Nature Micro submission. I have no further comments.

Response: We are quite grateful for Reviewer #3's positive feedback on our revision!

Reviewer #4 (Remarks to the Author):

We appreciate Reviewer #4's valuable time for reviewing our manuscript! Many thanks for your kind recognition of our efforts on addressing Reviewer #1's concerns! We fully agree Reviewer #4's opinions and have incorporated proper changes in the revised manuscript (highlighted in yellow). Please see our point-by-point answers below.

It appears that the authors have made a valiant effort in addressing the reviewer concerns by updating the manuscript and including additional control experiments. Nonetheless, I am concerned about major shortcomings that may have not been brought up or addressed in prior revisions.

Much of the premise of the paper relies on two critical mass spectrometry experiments: 1. The presumed discovery of a SUMO modification on a peptide containing NP-K65 (Supplementary data 2), and 2. A NP interactome dataset that identified the SUMO E3 protein TRIM28 (Supplementary Data 1). However, both these experiments produced puzzling results that are not exclusively consistent with a SUMO modification on K65:

1. For the identification of the SUMOylated NP peptide (line 114, Supplementary data 2), it is puzzling that the authors searched for di-G modified peptides because these indicate ubiquitination but not SUMOylation. The SUMOylation appendage for native SUMO would be much larger as there is no C-terminal Lys for trypsin cleavage. Hence, it is much more likely that K65 is significantly ubiquitinated (or ISGylated, which produces the same modification). Did the author check by Western blot for levels of ubiquitination or ISGylation? Both of those modification would leave the di-G appendage. It is important to clarify how Suppl. Data 2 was generated to identify the potential SUMOylation site? Was this NP overexpressed by itself, in which cell line? If in the presence of overexpression of a mutant SUMO form and/or E1/E2/E3 ligases, this needs to be described.

While the new mass spectrum in Supp. Fig. 1c showing the SUMOylated peptide is convincing, this was generated under overexpression conditions, and with mutant Q87R/Q88N SUMO, which adds a trypsin cleavage site to the C-terminus of SUMO. This spectrum still does not explain how the longer peptide (line 114) with di-G remnant could have been generated from a native SUMO modification.

Response: We thank the Reviewer for raising these important points!

(1) Initially, to identify the SUMOylation site of NP, poly-SUMOylated NP was enriched from HEK293T cells co-expressed with Flag-SARS2-NP and widely-used 6His-SUMO3 T90R mutant^{1, 2, 3, 4, 5} via Nickel beads and anti-Flag pulldown sequentially (original line 114, Supplementary data 2). We are sorry for the

incomplete information provided and confusion caused. The identified K65 GG remnant could be a possible SUMOylation site. Worth to explain, GG remnant on other lysine residues were not identified could be caused by low abundance without enrichment by K-ε-GG immunoaffinity beads. We agree that the experiment was not designed properly, and K65 GG remnant could also represent other types of modification under this setting.

Here, we designed a new workflow to map the NP SUMOylation site. HEK293T cells were co-transfected with SUMO3 WT/T90K and VSV-Flag-NP. SARS2-NP were then enriched by anti-Flag beads pull-down and subsequently digested by trypsin, which generates a mixture of modified and nonmodified peptides. Under these conditions, identification of modified peptides is challenging, because they represent a very small fraction of the total amount of peptides. Peptides containing GG-K were enriched by K-ε-GG immunoaffinity beads and analyzed by MS/MS (lines 112-115 in the revised manuscript, revised Supplementary Fig. 1c).

Revised Supplementary Fig. 1c, Schematic representation of the method used for mapping of NP SUMOylation site.

Unlike ubiquitin, ISG15, NEDD8 which leaves a small GG remnant on the modified lysine residue after trypsin digestion, SUMO leaves a larger signature that severely hampers the identification of modified peptides. Whereas trypsin digestion of protein modified by SUMO3 T90R generates peptide with a GG remnant easily identifiable by classical MS/MS (lines 115-119 in the revised manuscript, revised Fig. 1c, left) ^{1, 2, 3, 4, 5}.

Revised Fig. 1c, left, schematic representation of the signature tags left after trypsin digestion on peptides modified by SUMO3 WT/T90R.

In both SUMO3 WT-expressing cells and SUMO3 T90R-expressing cells, several GG-modified sites on SARS2-NP, i.e., K61, K266, K342, K347, K361, K375, K388, were identified by label-free quantification of the abundance of GG remnant. GG remnant on these sites is probably from endogenous ubiquitin, or ISG15, or NEDD8 modification. Only GG-modified K65 was identified specifically in SUMO3 T90R-expressing cells, indicating there is no endogenous GG remnant attached on K65, and K65 is the potential SUMOylation site of SARS2-NP (lines 119-126 in the revised manuscript, revised Fig. 1c, right).

Revised Fig. 1c, right, the mean intensity (label-free quantification) of GG-K sites of SARS2-NP identified by MS/MS in SUMO3 WT/T90K-expressing HEK293T cells.

Furthermore, previous data showed that SUMOylation of SARS2-NP completely diminished by substitution of K65 with an arginine (K65R) in HEK293T, HeLa, Vero E6, RAW264.7 and MEF cells (Fig. 1d, Supplementary Fig. 1e). In SUMOylation assay *in vitro*, SUMOylation of purified SARS2-NP was produced by purified TRIM28 protein, rather than inactive TRIM28 C651A. SUMOylation was still undetectable on purified SARS2-NP K65R, again indicating that TRIM28 SUMOylates the K65 residue (Fig. 6e).

(2) In Supplementary Fig. 1f, there was no evident difference of ubiquitination between NP WT and NP K65R. Meanwhile, ubiquitination was not observed when individual lysine residues were reintroduced to R65 of SARS2-NP K0 (SARS2-NP K0-R65K). It could be excluded that the possibility of K65 is ubiquitinated predominantly. The concern of SARS2-NP K0 construct was explained below.

To further address Reviewer's concern, we tested whether NP is ISGylated. As shown in figure below for Reviewer only, there is no SARS2-NP ISGylation could be observed clearly.

Fig for Reviewer only. ISGylation assay of SARS2-NP. IB of total lysates and anti-Flag IP from HEK293T cells transfected with plasmids expressing HA-ISG15, plus Flag-SARS2-NP WT/K65R plasmids or not as indicated.

2. The proteomics data in Supp. Data 1 needs further explanation. What is shown? Co-IP of NP? Where are non-transfected controls? How many replicates were analyzed and how were statistics calculated (none are presented). Several proteins, including TRIM28 are only identified by a single peptide. Standard in the field is to filter for at least 2 peptides per protein to ensure robust ID. The authors need to explain their rationale for less stringent IDs.

Overall, it seems surprising that the authors focused on TRIM28 because only 1 peptide was IDed in the MS experiments. TRIM21, an E3 with ubiquitination activity, is identified much more confidently as an interactor (13 peptides/18 PSMs). Why did the authors not follow up on this lead and focused instead on TRIM28?

Even though subsequent experiments provide convincing evidence that K65 can be SUMOylated by TRIM28, all these experiments are carried out under conditions of overexpression of various components (E3, SUMO, etc.). Unfortunately, these results do not distinguish the possibility that another NP-K65 PTM may be the predominant proteoform in a native infection context. The only experiment that tested for ubiquitination (Supplemental Figure 1e) used the NP-K0-R65K construct, which has all 29 other lysine residues mutated. As pointed out by reviewer 1, these mutations could influence NP stability and protein interactions, preventing the detection of native, more prominent PTMs on K65.

My subsequent concern is that the phenotypic experiments (interferon suppression, VSV infection) are all relying on the K65R NP as a non-SUMOylated form of NP. However, since the authors failed to convincingly show in the first sections of the paper that SUMOylation is the predominant PTM occurring on K65, I recommend a more cautious rewording of the data presented in Figures 3-5 & 7-8. Overall, the authors can conclude that the K65 (and R203K) NP modification sites (and SIM, etc.) are required for the observed phenotypes, but not necessarily SUMOylation.

Response: Many thanks for the comments! We hope our explanation of existed data could address Reviewer's concerns.

(1) **Supplementary Data 1 is the Mass Spec-identified proteomic interactome of SARS2-NP in the SARS-CoV-2-infected A549 cells stably expressing human ACE2 (lines 93-95). Non-infected cells were set as control. Each condition was triplicated. To find statistically significant differences between the two conditions (infection vs. non-infection), a Student's *t*-test with a permutation-based approach was applied with an FDR cut-off of 0.05 (lines 659-678 in the revised manuscript).**

(2) **Regarding the concern on single peptide selected, the SARS2-NP interactors were identified from SARS-CoV-2-infected cells, rather than SARS2-NP-overexpressed cells which may lead to promiscuous or otherwise irrelevant host proteins. Occasionally, enzymes form transient covalent bonds with substrates, lowering activation energy, many ubiquitination E3-substrate interactions for example⁶. To avoid missing identification of the transient or weak interactions, the interactors presented by single peptide were included for analysis. It has now been explained in the revised manuscript (lines 672-673 in the revised manuscript).**

In addition, we compared the SARS2-NP interactors between this study and article of Gordon, D.E. *et al.* which was mentioned below by Reviewer. Many interactors presented as single peptide in this study (highlighted by red) are also listed more than one by Gordon, D.E. *et al.*. Please see the figure in below item of response-to-comment.

(3) **In this manuscript, we focus on study of SARS2-NP SUMOylation. As discussed in the manuscript (lines 554-556). TRIM28 was chosen from the SARS2-NP interactors due to TRIM28 is a well-characterized SUMO E3 ligase^{7, 8, 9}. TRIM28 was demonstrated to be a NP interactor (Fig 6b, c, Supplementary Fig. 6a, b, c) via biochemistry experiments and responsible for SARS2-NP poly-SUMOylation via *in vivo/vitro* assay (please see item (4) below). Whereas TRIM21 is a ubiquitination E3 ligase, which could be responsible for SARS2-NP ubiquitination. In our earlier pilot experiment, TRIM21 was tested with no SUMO E3 activity for SARS2-NP SUMOylation (no further data shown here).**

(4) **Regarding the concern on components (SUMO, E3) overexpression. Introduction of 6His-SUMO3 is commonly used for SUMOylation assay *in vivo/vitro* as necessary. To demonstrate TRIM28 is responsible for SARS2-NP poly-SUMOylation *in vivo*, the TRIM28 level was manipulated by ectopic expression of TRIM28 or inactive TRIM28 C651A (Fig. 6d), or shRNA-mediated knockdown (Supplementary Fig. 6g, Fig. 6f), or Crispr/Cas9-mediated knockout (Supplementary Fig. 6d). In addition, poly-SUMOylation assay *in vitro* was also performed by using purified proteins (Fig. 6e).**

(5) **Poly-SUMOylation of SARS2-NP was clearly observed in SARS-CoV-2-infected 16HBE, A549-hACE2 and CaCo-2 cells, as well as lungs of hACE2-transgenic mice (Fig. 1b). To address Reviewer's concern on overexpression of various components and distinguishing the possibility of predominant NP K65 PTM in a native infection context, one option is using reverse genetics strategy to modify the SARS-CoV-2 genome sequence by substitution of the putative K65 with an arginine (SARS2-NP K65R). Then infecting cells with prototypical or modified SARS-CoV-2 for detecting and comparing PTM of interesting. However, it is not allowable to perform such experiment due to reverse genetic manipulation**

of SARS-CoV-2 is strictly prohibited in China and most other countries for the time being. Instead, HEK293T cells were infected by recombinant VSVs. There is no SUMOylation could be observed in VSV-NP K65R-infected cells. In addition, VSV-NP K65R-R203K showed gain of poly-SUMOylation based on K65R. the VSV-NP R203K, which contains two SUMOylation sites, showed higher poly-SUMOylation levels (revised Supplementary Fig. 7c, lines 386-390 in the revised manuscript).

Revised Supplementary Fig. 7c, SUMOylation assay. IB of total lysates and anti-SARS2-NP immunoprecipitates (IP) of cell lysates from VSV-NP WT/K65R/K65R-R203K/R203K-infected HEK293T cells.

(6) The K65 ubiquitination was excluded, and there is no SARS2-NP ISGylation could be observed clearly. However, there are many other PTMs can happen on lysine residue, such as URM1, FAT10, FUBI (totally more than 30 kinds of) modification, *etc.* We wish that Reviewer #4 could understand that it is almost not possible to exclude the possibility that other PTMs could not happen on the K65.

(7) In Supplementary Fig. 1f, there was no evident difference of ubiquitination between NP WT and NP K65R. Meanwhile, ubiquitination was not observed when individual lysine residues were reintroduced to R65 of SARS2-NP K0 (SARS2-NP K0-R65K). The possibility that K65 ubiquitination was excluded (copied from above).

Indeed, SARS2-NP K0 is an artificial construct. The technique to mutate all lysine residues to arginine (K0) had been adopted in many other studies on lysine residue associated post-translational modifications, such as SUMOylation of Sam68¹⁰, MAVS¹¹, α -kleisin¹², Methylation of Smyd3¹³, Acetylation of HMGCS2¹⁴, histone H4¹⁵, Ubiquitination of Cox12¹⁶, CD4¹⁷, ubiquitin itself^{18, 19, 20}, or in phase separation study of TDP43²¹, *etc.*

SARS2-NP K0 would have unpredicted influence on NP's functions. Thus, the two constructs, SARS2-NP K0/K0-R65K, were only used for SUMOylation and ubiquitination assay as a kind of control. They were not used for further studies.

To address Reviewer's concern, SARS2-NP K0-produced data (previous Fig. 1d, part of Fig. 1e, part of Supplementary Fig. 1e, Supplementary Fig.1f and Supplementary Fig. 7b) were removed. And the conclusion would not be changed without these data.

(8) To the reviewer's point on cautious rewording of the data presented. As another predominant PTM on NP K65 could not be excluded, the data presented in Fig. 3 has been reworded by following Reviewer's suggestion (lines 205, 226-228, 1153 in the revised manuscript).

In the later studies, it was demonstrated that the interaction between SUMO conjugation and SIM of SARS2-NP establishes intermolecular self-association of SARS2-NP, by which SARS2-NP is condensed into liquid-like droplets and high-molecule weight aggregates more efficiently (Fig. 4 and Supplementary Fig. 4). NP K65R (loss of poly-SUMOylation) and NP SIM1A (loss of SIM site) have similar effects on decreasing NP's functions of self-association (Supplementary Fig. 4j), aggregation (Supplementary Fig. 4k), RNA binding (Supplementary Fig. 4m, n), droplets formation without or with RNA (Supplementary Fig. 4p), and suppression of innate antiviral signaling (Supplementary Fig. 4r, Fig. 5 and Supplementary Fig. 5). Meanwhile, there was no additive or synergistic effect could be observed in double mutant (loss of both SUMOylation and SIM site).

R203K mutation of SARS2-NP happens in the prevalent and concerning variants. Compared to single poly-SUMOylation site (SARS2-NP WT or NP K65R-R203K), stronger poly-SUMOylation was observed in SARS2-NP R203K mutation, which leads to increased self-association, RNA binding, LLPS and immunosuppression of SARS2-NP (Fig. 7 and Supplementary Fig. 7). These results indicate that the intensity of observed phenotypes is relevant to the level of SARS2-NP poly-SUMOylation.

What is more supportive, some of the observed phenotypes are achieved by using eukaryote-purified poly-SUMOylated SARS2-NP or prokaryote-purified mimicking protein of SARS2-NP SUMOylation (Fig. 2a-h; Supplementary Fig. 2 a, b; Fig. 4b, c, e, f, h, i, k, l; Supplementary Fig. 4b, e, l, o, q; Fig. 7c-e and Supplementary Fig. 7j, k; *etc.*). Based on these results, we wish Reviewer could support our words using on these conclusions.

Other comments:

3. Even with modifications and reorganization, the figures remain extremely dense and are at times challenging to follow.

Response: We totally understand Reviewer's concern. On the condition that the narrative thread of the article would not be changed, some data (previous Fig. 2b; Fig. 5j; Fig. 6b, h, p) were transferred to supplementary figures (current supplementary Fig. 2c; Fig. 5c; Fig. 6b, h, p). In addition, Previous Supplementary Fig. 4c was redundant and removed.

4. When mapping the modifications on K203 in Supplementary Fig. 7a by mass spectrometry, the concern again is that this experiment relies on overexpression of the conjugating machinery and misses other potential di-G modifications that are also

possible.

Response: In response to Reviewer, we performed MS/MS to demonstrate K203 SUMOylation of SARS2-NP R203K mutant by following the illustration in revised Fig. 1c left and supplementary Fig. 1c. Result showed that GG remnant is attached on K203 of SARS2-NP R203K in SUMO3 T90R-expressing cells only, in addition to K65 (lines 383-386 in the revised manuscript, revised Supplementary Fig. 7a).

Revised Supplementary Fig. 7a, The mean intensity (label-free quantification) of GG-K sites of SARS2-NP identified by MS/MS in SUMO3 WT/T90K-expressing HEK293T cells.

In addition, SARS2-NP R203K was subjected to ubiquitination assay. As shown in figure below, Compared to SARS2-NP WT/K65R, the ubiquitination level was not increased in NP R203

K/K65R-R203K respectively, indicating there is lower level of or no ubiquitination happens on NP R203K (lines 392-394 in the revised manuscript, revised Supplementary Fig. 7f).

Revised Supplementary Fig. 7f, Ubiquitination assay. IB of total lysates and anti-Flag IP from HEK293T cells transfected with plasmids expressing HA-Ub, plus Flag-SARS2-NP WT/K65R/K65R-R203K/R203K as indicated.

Furthermore, there is still no SARS2-NP R203K ISGylation could be observed clearly, As shown in figure below for Reviewer only.

Fig. for Reviewer only, ISGylation assay of SARS2-NP R203K. IB of total lysates and anti-Flag IP from HEK293T cells transfected with plasmids expressing HA-ISG15, plus Flag-SARS2-NP WT/ R203K plasmids as indicated.

5. For new Supplementary Fig. 9b: Please clarify how this peptide binding assay was conducted. The IB shows the biotin signal at 55kDa, but the biotinylated peptide is likely much smaller. What is the signal here?

Response: Many thanks for the comment!

To determine NSIP-III binding to SARS2-NP in solution, prokaryote-purified SARS2-NP protein was incubated with NSIP-III labelled without or with biotin (100 nM each) at 4 °C for 1 h. Then 15 µl of streptavidin Sepharose were added to perform the standard pull-down of biotin-labelled NSIP-III. Thus, The IB signal around 55kDa is SARS2-NP that bound to NSIP-III. The information had been provided, please refer to lines 746-748.

6. For the NSPI-III peptide experiments in mice and Caco2 cells, the more appropriate vehicle control would be the unmodified HIV-TAT sequence, or an inactive peptide. Did the authors test those for activity?

Response: We fully agree Reviewer's comment. Yes, we had tested candidate peptides in Caco-2 cells in pilot experiments before, please see figure below for

Reviewer only. Compared to other 4 peptides, NSIP-III showed potent activity in repressing SARS-CoV-2 replication, as assessed by the expression of SARS-CoV-2 subgenomic RNA4-encoding *E* gene and viral titres. *IFNB1* mRNA expression was elevated following NSIP-III treatment.

Fig. for Reviewer only, CaCo-2 cells were pretreated PBS or NSIP- NSIP-I/II/III/IV/V (50 μ M) for 2 h and infected with SARS-CoV-2 for 12 h. Left, subgenomic RNA4-encoding *E* gene which was determined by qPCR to reflect SARS-CoV-2 RNA levels. Middle, the SARS-CoV-2 titres. Right, Normalized *IFNB1* mRNA. ND, not determined.

However, an inactive peptide was not included as a control in mice study. Now, we are trying to engineer the ACE2-tropism nanoparticles to load and deliver NSIP-III more effectively through cooperating with pharmacy expert. We would like to use an inactive peptide as a proper control in the following *in vitro/vivo* studies.

7. Experimental details are needed on the conditions and search settings for the mass spectrometry data analysis, in particular the search of modified peptides. Mass spectrometry raw data should be submitted to an online repository (e.g. PRIDE/ProteomeXchange).

Response:

We would like to thank the Reviewer for spotting this. The experimental details have now been described in the methods. Please refer to lines 659-687 in the revised manuscript.

The MS raw data were failed to uploaded to PRIDE partner repository with several attempts. Instead, they are available in the ProteomeXchange Consortium via the iProX partner repository with accession code (PXD046366). Share link: <https://www.iprox.cn/page/SSV024.html?url=1698395333464VeUF>. Password: 76BF. The MS raw data will be released to public when the paper associated with this iProX ID is published.

8. How do interactors compare to other NP interaction datasets, e.g. Gordon, D.E. et al. Nature 583, 459–468 (2020)?

Response: The article of Gordon, D.E. *et al.* cloned, strep-tagged and overexpressed 26 of the 29 SARS-CoV-2 proteins in HEK-293T cells. By using on-beads digestion and mass spectrometry, interactions between SARS-CoV-2 and human proteins were identified. Among them, 584 proteins were found to interact with SARS2-NP. In this study, 483 proteins were found to interact with SARS2-

NP in the SARS-CoV-2-infected A549 cells stably expressing human ACE2 by in-gel digestion and mass spectrometry.

In response to Reviewer, the SARS2-NP interactors were compared between this two studies. As shown in the Venn diagram below (for reviewer only), there are 267 interactors are overlapped.

Fig. for Reviewer only, Comparison of SARS2-NP interactors between Gordon *et al* and this study. Red indicates the interactors presented by single peptide in this study.

9. Explain how GO analysis in Fig. 1 and Supp. Fig. 1a differ. What do the arrows represent?

Response:

Fig. 1a is the scatter plot of enriched terms from gene ontology (GO) enrichment analysis of the SARS2-NP interactome.

Supp. Fig. 1a is the interactive graph of the enriched GO terms of SARS2-NP interactome. Each ontology term (called “class” in the field of ontologies) represents a functional characteristic. Terms can have relationships between them, such as one term being more specific than another term (also called “subclass”). The arrows represent the orientation relation between class and subclass. This information has been indicated in the revised legend of Fig. 1a (lines 1087) and Supplementary Fig. 1a (lines 1395-1398).

10. Line 201: HeLa data is shown in Fig. 2k - not 2e. Why the switch to Hela cells here?

Clarify whether these cells also overexpressed Ubc9 and SUMO3.

Response: Many thanks for the comment!

In this manuscript, HeLa cells were used for observing NP puncta formation *in cellulo*. We have demonstrated that SARS2-NP poly-SUMOylation can also be seen in HeLa cells (Supplementary Fig. 1e). Compared to HEK293T cells, HeLa cells can attach onto coverslip tightly, they are not easy to be washed away by buffers when performing immunofluorescent stain.

Previous Fig. 2e (current Fig. 2d) is the droplet formation of bacteria-purified proteins. SARS2-NP poly-SUMOylation was strongly enhanced by ectopic expression of SUMO-conjugating E2 enzyme Ubc9 in HEK293T cells (Supplementary Fig. 1b). To purify poly-SUMOylated SARS2-NP from eukaryotic cells as maximum as possible, we thus stably expressed Ubc9 and His-SUMO3 in HEK293F cells (Supplementary Fig. 2f). The cells with overexpressed SUMO3, or SUMO3 and Ubc9, had been indicated in corresponding figure and figure legends.

11. Line 240: Please explain the MST assays.

Response: The microscale thermophoresis (MST) assay is a technology for the biophysical analysis of interactions between biomolecules. The assay allows the measurement of interactions directly in the solution without the need for immobilization to a surface²². It was explained in the revised manuscript (lines 157-158).

12. In Supp. Fig. 6a in the input, NP is below the 35kDa MW marker, while in the IPs it appears slightly below the 55kDa marker. Is this an error?

Response: Yes, it was wrongly-labelled and now corrected. Many thanks for reminding!

13. Supplemental Fig. 8e: The MW marker shows that myc-TRIM28 is <55kDa, but the full-length protein is ~100kDa (see Fig. 8d). Is the MW marker incorrect, or was a TRIM28 truncation used for these experiments?

Response: Myc-TRIM28 in Supplemental Fig. 8e is a full-length construct. It was wrongly-labelled and corrected. Many thanks for reminding!

14. Line 441: Explain the HDock platform.

Response: HDock platform is a highly-integrated server for protein-protein docking (<http://hdock.phys.hust.edu.cn/>) through a hybrid strategy of template-based modeling and *ab initio* template-free docking. The server automatically incorporates the binding interface information from the PDB and/or user-input biological information like residue restraints and molecular size/shape information obtained from small-angle X-ray scattering (SAXS), supports both

amino acid sequence and structure inputs and uses an intrinsic scoring function for protein–protein interactions²³. It was briefly explained in the revised manuscript (lines 442-444).

15. Line 451: incomplete sentence

Response: Incomplete sentence was caused by wrongly using of punctuation. It should be a comma, rather than a full stop. This mistake has been corrected (lines 454-455 in the revised manuscript). Many thanks for reminding!

References

1. Matic I, *et al.* Site-Specific Identification of SUMO-2 Targets in Cells Reveals an Inverted SUMOylation Motif and a Hydrophobic Cluster SUMOylation Motif. *Molecular Cell* **39**, 641-652 (2010).
2. Tammsalu T, Matic I, Jaffray EG, Ibrahim AFM, Tatham MH, Hay RT. Proteome-wide identification of SUMO2 modification sites. *Sci Signal* **7**, rs2 (2014).
3. Impens F, Radoshevich L, Cossart P, Ribet D. Mapping of SUMO sites and analysis of SUMOylation changes induced by external stimuli. *Proceedings of the National Academy of Sciences* **111**, 12432-12437 (2014).
4. Sheng Z, *et al.* MS-based strategies for identification of protein SUMOylation modification. *ELECTROPHORESIS* **40**, 2877-2887 (2019).
5. Blomster HA, *et al.* In vivo identification of sumoylation sites by a signature tag and cysteine-targeted affinity purification. *J Biol Chem* **285**, 19324-19329 (2010).
6. Henneberg LT, Schulman BA. Decoding the messaging of the ubiquitin system using chemical and protein probes. *Cell Chem Biol* **28**, 889-902 (2021).
7. Ma X, *et al.* TRIM28 promotes HIV-1 latency by SUMOylating CDK9 and inhibiting P-TEFb. *eLife* **8**, e42426 (2019).
8. Bürck C, *et al.* KAP1Is a Host Restriction Factor That Promotes Human Adenovirus E1B-55K SUMO Modification. *Journal of virology* **90**, 930-946 (2015).
9. Li M, Xu X, Chang CW, Liu Y. TRIM28 functions as the SUMO E3 ligase for PCNA in prevention of transcription induced DNA breaks. *Proc Natl Acad Sci U S A* **117**, 23588-23596 (2020).
10. Babic I, Cherry E, Fujita DJ. SUMO modification of Sam68 enhances its ability to repress cyclin D1 expression and inhibits its ability to induce apoptosis. *Oncogene* **25**, 4955-4964 (2006).
11. Dai T, *et al.* MAVS deSUMOylation by SENP1 inhibits its aggregation and antagonizes IRF3 activation. *Nature Structural & Molecular Biology* **30**, 785-799 (2023).

12. McAleenan A, *et al.* SUMOylation of the α -Kleisin Subunit of Cohesin Is Required for DNA Damage-Induced Cohesion. *Current Biology* **22**, 1564-1575 (2012).
13. Van Aller GS, *et al.* Smyd3 regulates cancer cell phenotypes and catalyzes histone H4 lysine 5 methylation. *Epigenetics* **7**, 340-343 (2012).
14. Shimazu T, *et al.* SIRT3 Deacetylates Mitochondrial 3-Hydroxy-3-Methylglutaryl CoA Synthase 2 and Regulates Ketone Body Production. *Cell Metabolism* **12**, 654-661 (2010).
15. Smith CM, Gafken PR, Zhang Z, Gottschling DE, Smith JB, Smith DL. Mass spectrometric quantification of acetylation at specific lysines within the amino-terminal tail of histone H4. *Analytical Biochemistry* **316**, 23-33 (2003).
16. Kowalski L, Bragoszewski P, Khmelinskii A, Glow E, Knop M, Chacinska A. Determinants of the cytosolic turnover of mitochondrial intermembrane space proteins. *BMC Biology* **16**, 66 (2018).
17. Magadán JG, Pérez-Victoria FJ, Sougrat R, Ye Y, Strebel K, Bonifacino JS. Multilayered Mechanism of CD4 Downregulation by HIV-1 Vpu Involving Distinct ER Retention and ERAD Targeting Steps. *PLOS Pathogens* **6**, e1000869 (2010).
18. Zhang J, *et al.* ATM functions at the peroxisome to induce pexophagy in response to ROS. *Nature Cell Biology* **17**, 1259-1269 (2015).
19. Feng L, *et al.* MARCH3 negatively regulates IL-3-triggered inflammatory response by mediating K48-linked polyubiquitination and degradation of IL-3R α . *Signal Transduction and Targeted Therapy* **7**, 21 (2022).
20. Deng L, *et al.* Activation of the I κ B Kinase Complex by TRAF6 Requires a Dimeric Ubiquitin-Conjugating Enzyme Complex and a Unique Polyubiquitin Chain. *Cell* **103**, 351-361 (2000).
21. Schmidt HB, Barreau A, Rohatgi R. Phase separation-deficient TDP43 remains functional in splicing. *Nat Commun* **10**, 4890 (2019).
22. Wienken CJ, Baaske P, Rothbauer U, Braun D, Duhr S. Protein-binding assays in biological liquids using microscale thermophoresis. *Nat Commun* **1**, 100 (2010).
23. Yan Y, Tao H, He J, Huang S-Y. The HDOCK server for integrated protein-protein docking. *Nature Protocols* **15**, 1829-1852 (2020).

REVIEWERS' COMMENTS

Reviewer #4 (Remarks to the Author):

Overall, I am satisfied with the revisions that the authors made. Thank you for the extensive explanations addressing all my concerns. The update to the experiment in Fig. 1c-d comparing WT and T90R SUMOylation is especially appreciated. This is a much more convincing strategy to show that the K65 site on the nucleocapsid protein is indeed SUMOylated.

One minor suggestion is to include the test results of other candidate peptides against SARS-CoV-2 (for review only figure in rebuttal on p. 10) into Supplemental Fig. 9. This is valuable in vitro information to at least demonstrate that the effect is specific to the NSIP-III peptide.

Reviewer #4 (Remarks to the Author):

Overall, I am satisfied with the revisions that the authors made. Thank you for the extensive explanations addressing all my concerns. The update to the experiment in Fig. 1c-d comparing WT and T90R SUMOylation is especially appreciated. This is a much more convincing strategy to show that the K65 site on the nucleocapsid protein is indeed SUMOylated.

One minor suggestion is to include the test results of other candidate peptides against SARS-CoV-2 (for review only figure in rebuttal on p. 10) into Supplemental Fig. 9. This is valuable in vitro information to at least demonstrate that the effect is specific to the NSIP-III peptide.

Response: We are quite grateful for accepting our revision!

The test results of other candidate peptides against SARS-CoV-2 have been included in Supplemental Fig. 9j. Corresponding description has also been added (lines 478-479 in the revised manuscript). Many thanks for suggestion.